

**Aerosol pH Indicator and Organosulfate Detectability from Aerosol Mass Spectrometry**
**Measurements**
Melinda K. Schueneman[1], Benjamin A. Nault[1], Pedro Campuzano-Jost[1], Duseong S. Jo[1,2],
Douglas A. Day[1], Jason C. Schroder[1,*], Brett B. Palm[1], Alma Hodzic[2], Jack E. Dibb[3], and Jose L.
Jimenez[1]
[1] Department of Chemistry, and Cooperative Institute for Research in Environmental Sciences
(CIRES), University of Colorado, Boulder, CO, USA
[2] Atmospheric Chemistry Observations and Modeling, National Center for Atmospheric
Research, Boulder, CO 80301, USA
[3] Earth Systems Research Center, Institute for the Study of Earth, Oceans, and Space,
University of New Hampshire, USA
* Now at: Air Pollution Control Division, Colorado Department of Public Health and the
Environment, Denver, CO, USA
Correspondence: jose.jimenez@colorado.edu



**Abstract**
Aerosol sulfate is a major component of submicron particulate matter (PM$_1$). Sulfate can be
present as inorganic (mainly ammonium sulfate, AS) or organic sulfate (OS). Although OS are
thought to be a smaller fraction of total sulfate in most cases, recent literature argues that this
may not be the case in more polluted environments. Aerodyne Aerosol Mass Spectrometers
(AMS) measure total submicron sulfate, but it has been difficult to apportion AS vs. OS as the
detected ion fragments are similar. Recently, two new methods have been proposed to quantify
OS separately from AS with AMS data. We use observations collected during several airborne
field campaigns covering a wide range of sources and airmass ages (spanning the continental US,
marine remote troposphere, and Korea) and
targeted laboratory experiments to investigate
the performance and validity of the proposed
OS methods. Four chemical regimes are
defined to categorize the factors impacting
sulfate fragmentation (Fig. shown in abstract).
In polluted areas with high ammonium nitrate
concentrations and in remote areas with high
aerosol acidity, the decomposition and
fragmentation of sulfate in the AMS is
influenced by multiple complex effects, and
estimation of OS does not seem possible with
current methods. In regions with lower acidity (pH>0) and ammonium nitrate (fraction<0.3), the
proposed OS methods might be more reliable, although application of these methods often
produced nonsensical results. However, the fragmentation of ambient neutralized sulfate varies
somewhat within studies, adding uncertainty, possibly due to variations in the effect of organics.
Under highly acidic conditions, sulfate fragment ratios show a clear relationship with acidity (pH
and ammonium balance). The measured ammonium balance (and to a lesser extent, the
H$_y$SO$_x^+$/SO$_x^+$ AMS ratio) is a promising indicator for rapid estimation of aerosol pH < 0,
including when gas-phase NH$_3$ and HNO$_3$ are not available. These results allow an improved
understanding of important intensive properties of ambient aerosols.



## Introduction

PM$_1$, or submicron aerosols, have important impacts on visibility, climate, and

environmental and human health (Dockery et al., 1996; Lighty et al., 2000; Lohmann et al.,

2004; IPCC, 2013). In order to quantify the impacts of PM$_1$, and their evolution with changes in

emissions, chemistry, and climate, PM$_1$ sources, chemistry, and composition must be understood.

Field measurements are critical to that goal, and one tool used extensively in field studies since

the early 2000s is the Aerodyne Aerosol Mass Spectrometer (AMS) and more recently its

simplified version, the Aerosol Chemical Speciation Monitor (ACSM) (Jayne et al., 2000;

DeCarlo et al., 2006; Canagaratna et al., 2007; Ng et al., 2011a)). The AMS typically quantifies

the chemical composition and size distribution of sulfate, nitrate, organic aerosol (OA),

ammonium, and chloride (Jayne et al., 2000; DeCarlo et al., 2006; Canagaratna et al., 2007;

Jimenez et al., 2009)

Within the AMS, particles are vaporized, leading to some thermal decomposition (e.g.,

(Docherty et al., 2015) and then ionized via 70 eV electron ionization, which leads to substantial

fragmentation of the molecular ions. Despite or perhaps because of the substantial (and

reproducible) decomposition and fragmentation, the relative signals of different AMS fragments

have been found to be indicative of different chemical species in the aerosol. These include the

presence of inorganic vs. organic nitrates (Farmer et al., 2010; Fry et al., 2013), and of several

source and composition characteristics of organic aerosols (Alfarra et al., 2004; Zhang et al.,

2004a; Cubison et al., 2011; Ng et al., 2011b; Hu et al., 2015). In contrast to nitrates,

deconvolving inorganic vs. organic sulfates is thought to be more difficult, as the fragmentation

pattern for one atmospherically relevant organosulfate (OS) was similar to those of inorganic



sulfates (mainly ammonium-sulfate salts, AS) in an early study, with minimal C-S-containing
fragments (Farmer et al., 2010). Until recently, most studies have shown that the OS molar
fraction ($OS_f$ = OS / (AS + OS), calculated using only the sulfate moiety of the molecules)
typically makes a small (~1-10%) contribution to total sulfate in $PM_1$ (e.g. (Tolocka and Turpin,
2012; Hu et al., 2015; Liao et al., 2015; Riva et al., 2016, 2019a)). However, for biogenic areas
$OS_f$ is predicted to increase substantially in the future (Riva et al., 2019b). Another important
recent subject of debate is the missing sulfate production in haze events in China (Wang et al.,
2014; Zheng et al., 2014; Li et al., 2017), which some studies have attributed to a major
contribution of OS (e.g., (Song et al., 2019)). It is also important to quantify OS in order to
understand  the chemistry of aerosol formation and aging (Surratt et al., 2007, 2008; Song et al.,
2019), which impacts the ability to understand how sulfate may influence various $PM_1$ properties
and processes (e.g., gas uptake, aqueous reactions). Finally, accurate AS concentrations are
needed to quantify the inorganic:organic ratio (to predict the hygroscopicity of $PM_1$, which
impacts satellite and model interpretation) and to estimate aerosol pH and liquid water content
from thermodynamic models, as it is currently still not possible to measure the aerosol pH in the
field in-situ (Hennigan et al., 2015; Guo et al., 2016; Craig et al., 2018; Pye et al., 2019).

Recent AMS work has attempted to quantify $OS_f$ from the measured individual sulfate

ion signals. The vaporization and ionization of AS and OS in the AMS produces almost
exclusively "inorganic" ion fragments, the major ones quantified being $SO^+$, $SO_2^+$, $SO_3^+$, $HSO_3^+$,
and $H_2SO_4^+$ (Farmer et al., 2010). Note that these are the ions detected in the AMS (following
ionization/decomposition), and not the ions present in the aerosols (discussed in Sect. 3.2 and
shown in Fig. 2C). However, recent laboratory studies with many OS standards have found





reproducible differences in the fragmentation of AS vs OS (Chen et al., 2019). That study
proposed a method using the unique AS ion fragments ($H_2SO_4^+$ and $HSO_3^+$) divided by the total
sulfate signal ($H_2SO_4^+ + HSO_3^+ + SO_3^+ + SO_2^+ + SO^+$) to apportion OS, AS, and methylsulfonic acid
(MSA, an organosulfur compound, but not an organosulfate) in field datasets. It is important to
note that MSA can be directly measured with the (HR-)AMS (Phinney et al., 2006; Zorn et al.,
2008; Huang et al., 2017; Hodshire et al., 2019), so quantification of MSA with the method in
Chen et al. is not necessary. From this method, an average OS mass concentration ($C_{OS}$) of 0.12
$\mu g\ m^{-3}$ was estimated for the SOAS ground campaign in rural Alabama (Carlton et al., 2018),
with $OS_f \sim 4\%$ (Chen et al., 2019). That estimate is consistent with others for that site and region
(Hu et al., 2015; Liao et al., 2015). An alternative method to estimate $OS_f$ based on the same
principle was proposed by Song et al. (2019) using the observed AMS $SO^+/H_ySO_x^+$ and
$SO_2^+/H_ySO_x^+$. These authors reported $OS_f \sim 17\% \pm 7\%$ (which corresponds to $[OS] \sim 5\text{-}10\ \mu g\ m^{-3}$)
during winter haze episodes in China, based on their method. A recent study (Dovrou et al.,
2019) investigated mixtures of sodium sulfate and sodium hydroxymethanesulfonate (HMS);
however, they found that HMS cannot be distinguished from AMS ions alone, due to the
complex ambient aerosol mixture containing organic sulfates, and inorganic sulfates, which all,
in part, produce the same sulfate fragments as HMS.

Another important and related analytical challenge is online quantification or estimation

of ambient aerosol acidity from real-time measurements, e.g. during field campaigns. So far,
online aerosol pH measurements have only been performed in the laboratory (Rindelaub et al.,
2016; Craig et al., 2018). Aerosol acidity is important because it impacts human health by
decreasing lung function (Raizenne et al., 1996), and strongly impacts the equilibria and kinetics





of a very large number of atmospheric physical and chemical processes (Jang et al., 2002;
Meskhidze et al., 2003; Anon, 2007; Thornton et al., 2008; Bertram and Thornton, 2009; Gaston
et al., 2014; Ackendorf et al., 2017; Guo et al., 2017; Losey et al., 2018). In addition, the
deposition of acidic particles leads to damage to terrestrial and freshwater ecosystems, i.e. "acid
rain" or more properly acid deposition (Schindler, 1988; Johnson et al., 2008). Currently, the
state-of-the art technique to quantify aerosol acidity for field data is to run an inorganic aerosol
thermodynamic model that includes the measured particle and gas inorganic concentrations, as
well as temperature and humidity. The Extended Aerosol Inorganics Model (E-AIM) (Clegg et
al., 1998a, 2003; Wexler and Clegg, 2002) is generally considered as the reference model (Pye et
al., 2019). ISORROPIA-II (Nenes et al., 1999; Fountoukis and Nenes, 2007) is a faster model
utilizing look-up tables to calculate aerosol liquid water content (and thus is frequently used as
part of chemical transport models) at the expense of some accuracy at different RH levels (Pye et
al., 2019). In general, these thermodynamic models are thought to perform best for pH
estimation when gas-phase measurements of $NH_3$ and/or $HNO_3$ are used in the calculations, and
to perform less well when run only with aerosol measurements (Guo et al., 2015; Hennigan et
al., 2015; Song et al., 2018).

There has been an ongoing debate about the potential relationship between the inorganic

cation/anion charge ratio (commonly referred to as "ammonium balance", see Eq. (4)) and
aerosol acidity. Ammonia gas and its particle phase equivalent (ammonium) are the dominant
bases in the atmosphere (Dentener and Crutzen, 1994). As the most important base in $PM_1$, a
deficit of $NH_4^+$ vs. dominant $PM_1$ anions, $SO_4^{2-}$ and $NO_3^-$ (Jimenez et al., 2009), is indicative of
the concentration of $H^+$, since the particles are (nearly) electrically neutral. Thus, in the absence



of substantial non-volatile cations (e.g. $Na^+$, $K^+$) ammonium balance is an indicator of aerosol
acidity. Ammonium balance has been shown to correlate well with pH under certain conditions,
specifically, when using daily averaged temperature and relative humidity (Zhang et al., 2007a),
but has been criticized as being a poor surrogate of pH under other conditions (Hennigan et al.,
2015). In particular, ammonium balance can be a poor surrogate of pH because changes in T and
RH impact the aerosol liquid water in the diurnal cycle (Zhang et al., 2007a). This is especially
important in the boundary layer where almost all past pH quantification has been carried out
(Pye et al., 2019), compared to the lower diurnal variance of T and RH in the free and upper
troposphere. Many field studies do not include measurements of $NH_3$ or $HNO_3$, sticky species
present at low concentrations and thus not routinely measured, limiting the ability to calculate
aerosol pH (Hennigan et al., 2015). A more direct estimate of aerosol acidity using only ambient
particle data is highly desirable.

Here, we analyze sulfate ion fragment data from laboratory and ambient AMS

observations, spanning multiple aircraft campaigns with a routinely calibrated AMS response to
AS, and across a wide range of chemical and meteorological environments. We use this large
dataset to test the applicability of recently published methods to partition AS and OS. We
investigate the feasibility of estimating pH based on AMS data; as well as the regions of
chemical space where the different estimation methods may work. Finally, we provide a physical
interpretation for sulfate fragmentation in the AMS.

**2 Methods**
**2.1 Airborne Campaigns**



Sulfate fragmentation data was obtained using an Aerodyne High-Resolution

Time-of-Flight Aerosol Mass Spectrometer (AMS) (Aerodyne Research Inc., Billerica, MA,
USA; (DeCarlo et al., 2006)). The ambient data used here are from aircraft observations from the
following campaigns (Table 1): DC3 (Barth et al., 2015), SEAC⁴RS (Toon et al., 2016),
WINTER (Schroder et al., 2018), KORUS-AQ (Nault et al., 2018), and ATom-1 and ATom-2
(Guo et al., 2020; Hodzic et al., 2020)). Flight paths for all six campaigns are shown in Fig. S1.
These campaigns span polluted urban, partially polluted biogenic, biomass burning smoke, rural,
and remote regions of the atmosphere. DC3 sampled continental / rural conditions with diffuse
pollution and some biomass burning events. WINTER and KORUS-AQ were airborne
campaigns that focused on urbanized regions (although from different regions and times of year
(Table 1)); therefore, the campaigns had appreciable mass concentrations of ammonium nitrate
due to anthropogenic emissions of $NO_x$ and the subsequent production of $HNO_3$ that partitions
into the aerosol with ammonia (Seinfeld and Pandis, 2006). SEAC⁴RS focused on regional
background chemistry of the continental United States, which included impacts from biomass
burning, biogenic, and pollution emissions, and upper tropospheric chemistry impacted by
convection. Finally, ATom-1 and ATom-2 sampled the remote Pacific and Atlantic basins with
continuous full vertical profiling, in order to study the composition of the remote marine
atmosphere, impacted by long range transported chemical species and marine emissions, and far
from anthropogenic sources. Not all campaigns are usable for all the analyses in this paper,
depending on the quality and completeness of the data. Table 1 indicates which campaigns were
usable for each analysis.


## 2.2 High-Resolution Time-of-Flight Aerosol Mass Spectrometer


The highly customized University of Colorado-Boulder aircraft AMS was used in all
campaigns and has been described elsewhere (DeCarlo et al., 2008; Dunlea et al., 2009; Nault et
al., 2018; Schroder et al., 2018; Guo et al., 2020), so only details relevant to this study are
summarized here. Ambient air is drawn through a National Center for Atmospheric Research
(NCAR) High-Performance Instrumented Airborne Platform for Environmental Research
Modular Inlet (HIMIL: (Stith et al., 2009)) with a constant standard flow rate of 9 L min$^{-1}$, and
all data is reported at a constant standard temperature (T = 273 K) and pressure (P = 1013 hPa).
The sampled air enters a pressure controlled inlet (Bahreini et al., 2008) and is then introduced
into an aerodynamic focusing lens (Liu et al., 1995; Zhang et al., 2004b). Particles then impact
onto an inverted cone porous tungsten "standard" vaporizer, operated at ~ 600 °C under high
vacuum. The standard vaporizer is used in this study. A "capture vaporizer" has been recently
demonstrated, it leads to more thermal decomposition while still retaining similar (although
noisier) fragment information (Hu et al., 2017a; Zheng et al., 2020), but it is not used here.
Non-refractory species, those that evaporate in less than a few seconds (such as sulfate, nitrate,
ammonium, and organic material), are subsequently ionized by 70 eV electrons. Some refractory
and semi-refractory species such as sea-salt, lead and potassium can be detected by the AMS in
some cases (Lee et al., 2010; Salcedo et al., 2010; Ovadnevaite et al., 2012; Hodzic et al., 2020)).
A cryopump reduces background in the ionizer by orders of magnitude during the flights, leading
to low detection limits, in particular for NH$_4$, which is critical for acidity quantification in the
remote troposphere. Data was taken at 1 Hz, but was processed at both 1 Hz and 1 minute
resolution, and the latter product is primarily used here due to higher signal-to-noise ratios. The


one minute datasets were further filtered by removing points where the sulfate signal was below
three times its detection limit. Detection limits were estimated continuously via the methods of
Drewnick et al. (2009), and confirmed with frequent in-flight filter blanks. For the laboratory
studies, everything was kept the same as on the aircraft other than no use of the HIMIL aircraft
inlet. Data was processed and analyzed with the standard Squirrel and PIKA ToF-AMS data
analysis software packages within Igor Pro 7 (Wavemetrics) (DeCarlo et al., 2006; Sueper,

2018).

One important parameter for AMS quantification is collection efficiency (CE). CE is the

probability that a particle entering the AMS is detected. It is affected by several particle
properties (Huffman et al., 2005), the most important being particle bounce off the vaporizer
without detection (Middlebrook et al., 2012). Bounce is controlled by particle phase (Quinn et
al., 2006; Matthew et al., 2008), and is estimated for ambient particles based on their ammonium
balance (acidity) and ammonium nitrate content (Middlebrook et al., 2012). This
parameterization performs well for ambient particles (Middlebrook et al., 2012; Hu et al., 2017a,
2020; Guo et al., 2020). Still, potential variability in CE that is not perfectly captured by the
parameterization contributes a major fraction of the AMS uncertainty for ambient particle
analysis (Bahreini et al., 2009). Alternative methods to estimate ambient CE for ambient
particles are of interest.

**2.3 Quantification of OS/AS using Literature Methods**

Two methods have been proposed to quantify OS contribution to total sulfate using AMS

sulfate ion fragment fractions. The first method uses different sulfate ions to attribute measured



total sulfate to either OS, AS, or methanesulfonic acid (MSA). Due to the structure of OS, only
non-hydrogenated sulfate ions, i.e., $SO^+$, $SO_2^+$ and $SO_3^+$, are produced in the AMS for OS. AS
does produce hydrogenated sulfate ions, i.e., $H_2SO_4^+$ and $HSO_3^+$, as well as the same
non-hydrogenated sulfate ions produced by OS. Chen et al. (2019) proposed a "triangle method"
to estimate these two species and MSA, based on the observed fragments. Note that mineral
sulfates such as sodium sulfate fragment similarly to OS, and thus these methods need to be
interpreted differently in regions with significant submicron mineral sulfates. MSA calibrations
show variability for the fragments (Chen et al., 2019), and were not performed for all the studies
in this work. Since MSA can be quantified without using the sulfate fragments, here we apply
this method to estimate the fractions of OS and AS by using a one dimensional version of the
triangle (i.e. just the hypotenuse connecting pure OS to pure AS).  An alternative method is
based on the same assumptions, but uses different equations to quantify the relative
concentration of OS (Song et al., 2019).

Both literature methods for deconvolving sulfate as OS and AS assume that the main

factor impacting sulfate fragmentation in the AMS is sulfate structure (OS, AS, or MSA). Chen
et al. (2019) briefly mention that acidity can impact sulfate fragmentation, but this effect has not
been studied and quantified. In addition, Chen et al. (2019) used pure standards to quantify the
AMS fragmentation of different species, but did not explore potential matrix effects in AMS
fragments which could impact internally mixed ambient particles.

**2.4 Quantification of the AMS Sulfate Fragment Ratios**





To compare our field data to that analyzed in Chen et al. (2019) we use the variables
defined in that study, $fH_2SO_4^+$ and $fHSO_3^+$ and define the normalized $nfH_2SO_4^+$ and $nfHSO_3^+$
(normalized to the values of $fH_2SO_4^+$ and $fHSO_3^+$ for pure AS) :

$$fH_2SO_4^+ = \frac{[H_2SO_4^+]}{[H_2SO_4^+]+[HSO_3^+]+[SO_3^+]+[SO_2^+]+[SO^+]}$$  Eq. 1


$$nfH_2SO_4^+ = \frac{fH_2SO_4^+}{fH_2SO_4^+ \, (pure \, AS)}$$  Eq. 2


$$fHSO_3^+ = \frac{[HSO_3^+]}{[H_2SO_4^+]+[HSO_3^+]+[SO_3^+]+[SO_2^+]+[SO^+]}$$  Eq. 3


$$nfHSO_3^+ = \frac{fHSO_3^+}{fHSO_3^+ \, (pure \, AS)}$$  Eq. 4


It should be noted that while that study includes methanesulfonic acid (MSA) data, the impact of
MSA on $fH_2SO_4^+$ and $fHSO_3^+$ is minimal for the ATom campaigns (see Fig. S2). Additionally,
one study over the Western United States (representing a rural, continental region) observed
MSA concentrations of ~50 ng m$^{-3}$ (Sorooshian et al., 2015), which results in a very small
deviation in the Chen triangle and can hence be neglected for the purposes of this work. All



variables were normalized to the values of the same variables for pure AS calibrations
(conducted during each field experiment) in order to eliminate some of the spread in the sulfate
ions that is likely due to instrument-to-instrument or instrument-in-time variability (Fry et al.,
2013; Chen et al., 2019) (Fig. S3) . We also define a new AMS sulfate ion ratio, $H_ySO_x^+/SO_x^+$, as
and create the normalized $nH_ySO_x^+/SO_x^+$ to reduce the influence of instrument-instrument
variability:

$$H_ySO_x^+/SO_x^+ = \frac{[H_ySO_x^+]}{[SO_x^+]} = \frac{[H_2SO_4^+]+[HSO_3^+]}{[SO_3^+]+[SO_2^+]+[SO^+]} \qquad \text{Eq. 5}$$


$$nH_ySO_x^+/SO_x^+ = \frac{H_ySO_x^+/SO_x^+}{H_ySO_x^+/SO_x^+ \, (pure \, AS)} \qquad \text{Eq. 6}$$


The submicron aerosol molar Ammonium Balance ($NH_{4\_bal}$) is calculated as:

$$NH_{4\_bal} = \frac{[NH_4]/18}{([SO_4]/48)+([NO_3]/62)+([Chl]/35)} \qquad \text{Eq. 7}$$


The concentration of non-refractory chloride is only included for non-remote campaigns
(KORUS-AQ, WINTER, and SEAC[4]RS), since it was negligible for others and strongly
impacted by seasalt in the marine boundary layer. The fraction of ammonium nitrate in the
particle phase (ammonium nitrate mass fraction, $AN_f$) (by mass):

$$AN_f = \frac{(80 \div 62) \times [Inorganic \, NO_3]}{[NO_3]+[SO_4]+[NH_4]+[Chl]+[Org]} \qquad \text{Eq. 8}$$




The fraction of total AMS aerosol mass comprised of OA ($OA_f$) is:

$$OA_f = \frac{[Org]}{[NO_3]+[SO_4]+[NH_4]+[Chl]+[Org]}$$   Eq. 9


The sulfate equivalent concentration of OS in the Song et al. (2019) paper is calculated as:

$$C_{OS} = M_{SO_4^{2-}}[\frac{SO_{obs}^+ - R_{cd,SO^+/H_ySO_x^+} \cdot H_ySO_{x,obs}^+}{M_{SO^+}} + \frac{SO_{2,obs}^+ - R_{cd,SO_2^+/H_ySO_x^+} \cdot H_ySO_{x,obs}^+}{M_{SO_2^+}}]$$   Eq. 10


where "cd" stands for "clean and dry". Clean and dry conditions are defined in Song et al. (2019)
as ambient data points where $PM_1 = 10$ μg m$^{-3}$ and RH = 30%. Clean and dry conditions are
assumed to represent nearly pure AS. M is for the molar mass of the different sulfate ions, and
"obs" represents the ambient data for specific sulfate fragments. $H_ySO_x^+$ is defined in Song et al.
(2019) as ($SO_3^+ + HSO_3^+ + H_2SO_4^+$). For the Chen method, the $C_{OS}$ is defined based on the AS
normalized $nfH_2SO_4^+$ values:

$$C_{OS} = [SO_4] - nfH_2SO_4^+ * [SO_4]$$   Eq. 11


$OS_f$, the fraction of OS:total sulfate is defined as:

$$OS_f = \frac{C_{OS}}{[SO_4]}$$   Eq. 12





Where $C_{OS}$ is calculated from Eq. (10) or Eq. (11).

### 2.5 Laboratory Experiments

294  As ambient aerosols contain mixtures of chemical species, we investigated if matrix

effects may impact the fragmentation of sulfate species. Different solution mixtures, composed
of various amounts of AS (Certified ACS, 99.7% purity) and ammonium nitrate (AN) (Certified
ACS, 99.9% purity) in water (Milli-Q grade (R > 19 MOhms)) were atomized to generate
particles and size selected using a Differential Mobility Analyzer (DMA) (TSI Model 3081),
analyzed with a Condensation Particle Counter (CPC) (Model 3775), and electrostatic classifier
(Model 3080), for mobility diameters between 350-400 nm. We investigated AS/AN mixtures,
ranging from $AN_f = 0\%$ to 95%.

302  In order to assess effects on the sulfate fragmentation from mixing with OA, chamber

experiments, where different types of SOA were formed by gas-phase reactions and
condensation onto AS seeds, were investigated. SOA was formed from alkanol and toluene
photooxidation under high-$NO_x$ conditions (Liu et al., 2019), as well as $\Delta$-3-carene and $\alpha$-pinene
reactions with nitrate radicals (Kang et al., 2016) Experiments were initiated with 100% AS in a
dry chamber (RH < 5%; ~ 298 K) followed by either rapid, gradual, or stepwise increases of
SOA until a maximum OA/(OA+AS) ratio of ~ 70% was reached. Aerosol composition was
monitored by AMS and size distributions were monitored with a scanning mobility particle sizer
(SMPS, TSI). RIE of sulfate was directly calibrated with pure ammonium sulfate, while RIE *
CE of the SOA produced was estimated by comparison to the SMPS integrated volume, together
with OA density estimated from the AMS-derived elemental ratios per Kuwata et al. (2012)), in



order to accurately quantify OA/(OA+AS). Humid experiments were not considered here due to
the potential of forming organosulfates.

*2.6 E-AIM Thermodynamic Model for pH Estimation*

Aerosol pH was estimated using the Extended Aerosol Inorganic Model (E-AIM) Model

IV (Clegg et al., 1998b; Massucci et al., 1999; Wexler and Clegg, 2002). We input into the model
the total nitrate (gas and particle phase), particle phase ammonium and sulfate, and ambient T
and RH to calculate aerosol liquid water and aerosol pH. Model IV is not run with chloride ions,
as their concentrations were very low, and including chloride limits the model to temperatures $\geq$
263 K (Friese and Ebel, 2010), which would greatly limit the analysis of pH for WINTER,
ATom-1, and ATom-2. Also, including chloride precludes running the model under
supersaturated solution conditions, which is a closer approximation of ambient aerosol (Pye et
al., 2019). The model was run in the "forward mode," meaning that total nitrate (gas-phase
$HNO_3$ plus particle-phase total $NO_3^-$), sulfate, ammonium, relative humidity (calculated
according to the parameterization of Murphy and Koop (2005) which is critical for upper
tropospheric conditions), and temperature were input into the model. All aerosol mass
concentrations were from the CU AMS. $HNO_3(g)$ was measured by the California Institute of
Technology chemical ionization mass spectrometer (CIT-CIMS) (Crounse et al., 2006), which
was flown in all of these missions (excluding WINTER, where the UW-CIMS was used for the
HNO3 measurements) (Lee et al., 2014, 2018). Results are generally similar when using the
SAGA mist chamber measurement for total nitrate (Nault et al., 2020). The forward mode is less
sensitive to uncertainties in measurements than the "reverse mode," which only uses particle



composition and T/RH as inputs (Hennigan et al., 2015). Also, due to lack of $NH_3(g)$
measurements, the model was run iteratively until convergence in modeled $NH_3$ occurred,
similar to Guo et al. (2016). Performance for modeled pH was investigated by comparing
model-calculated $HNO_3$ and $NO_3^-$ to measurements, as the partitioning of nitrate between gas-
and particle-phase is sensitive to pH under acidic conditions (Guo et al., 2016). For all
campaigns included herein (DC3, WINTER, SEAC[4]RS, KORUS-AQ, ATom-1, and ATom-2),
the slopes of $HNO_3$ (measured vs. predicted) are within the uncertainty of the measurements;
with good correlations (SI Fig. S4). For $NO_3^-$, the slopes are within the measurement uncertainty
for five of the six campaigns. For ATom-2, the $NO_3^-$ slopes were low; however, for this
campaign, the measured $NO_3^-$ mass concentrations were extremely low (mean = 0.02 μg sm$^{-3}$),
and the pH was also very low (mean = - 0.5), leading to very little $NO_3^-$ in the aerosol phase (see
SI Fig. S4).

In addition, other bases present in the atmosphere (such as amines) were examined. Prior

studies have shown that amines were less than a maximum concentration of 30 ng m$^{-3}$ at the
ocean surface (Gibb et al., 1999; Facchini et al., 2008; Müller et al., 2009; Frossard et al., 2014;
van Pinxteren et al., 2015; Youn et al., 2015). Another study found that amine mass
concentration dropped off quickly with altitude to concentrations less than 10 ng m$^{-3}$ at the
altitude that the DC-8 flew over marine surfaces (Sorooshian et al., 2009). As the one minute
detection limit for the AMS data for amines is typically 10 ng m$^{-3}$, we expect the amine signal to
generally be below the limit of detection, and thus outside of our quantification capabilities. This
was observed for AMS data from the ATom campaigns, using characteristic ions identified in
past studies (Murphy et al., 2007; Ge et al., 2014). It was found that amine ions cannot be



distinguished from background for many ATom flights. Only during one flight in ATom-1, we
observed an amine signal ($C_2H_6N^+$ $m/z$ = 44) above the background (see SI Fig. S5). During this
flight, amines (from the contribution of $CH_4N$, $C_2H_6N$, and $C_3H_8N$) only accounted for 0.7 ng
m$^{-3}$ of aerosol, whereas ammonium accounted for 19 ng m$^{-3}$. Amines can produce the same
fragments as ammonium, but this is only the case for a few percent of the amine fragments (Ge
et al., 2014). In this case, the ammonium concentration is 25 times that of the amines.  Since
amines were even lower during other flights, we assume the effect of amines to the pH
calculation is very small and can be ignored for E-AIM calculations.

## 2.7 GEOS-Chem Model

We used a global chemical transport model (GEOS-Chem 12.6.1,

doi:10.5281/zenodo.3520966; (Bey et al., 2001)) to investigate modeled global distributions of
ammonium nitrate mass fraction (AN$_f$) and aerosol pH across different regions. GEOS-Chem
was driven by assimilated meteorological fields from the Modern-Era Retrospective analysis for
Research and Applications version 2 (MERRA2) (Gelaro et al., 2017) for the year of 2010. The
simulation was conducted at 2˚ (latitude) × 2.5 (longitude) with 47 vertical layers up to 0.01 hPa
and ~30 layers under 200 hPa. We used the Community Emissions Data System (CEDS)
inventory for global anthropogenic emissions (Hoesly et al., 2018) and the global fire emissions
database version 4 (GFED4) for biomass burning emissions (Giglio et al., 2013). Aerosol pH and
gas-particle partitioning of inorganic aerosols were calculated online using the ISORROPIA-II
model within GEOS-Chem (Fountoukis and Nenes, 2007; Pye et al., 2009). Similar to Jo et al.,
(2019) sea salt aerosol was excluded from pH calculations based on a better agreement with the



observationally-constrained pH values as suggested by Nault et al. (2020). GEOS-Chem includes
sea salt aerosol in ISORROPIA calculation but we excluded sea salt aerosol based on Nault et al.
(2020). Oceanic $NH_x$ emissions were also included in this model based on recent work (Paulot et
al., 2015; Nault et al., 2020).

**3 Results and Discussion**
**3.1 Lab quantification of AMS data**

Application of the one dimensional Chen method to laboratory data is shown in Fig. 1.

Data are expected to lie inside the triangular region, and be apportioned depending on the
relative distance to the three vertices. For example, data lying at [0.5,0.5] on the line between the
OS and AS points would represent a sample with ~50% OS and ~50% AS. If data clusters
around the [1,1] point where pure AS resides, all of the sulfate is attributed to AS.

The effect of internally mixed ammonium nitrate (AN) is shown in Fig. 1A. For mixtures

containing $AN_f < 50\%$,  data centers around the pure AS point in the Chen triangle. When $AN_f$ is
increased past 0.50, there is an increase in both $nfH_ySO_x^+$ ions. For example, an aerosol with $AN_f$
= 0.75 results in $OS_f$= - 11% with the Chen method, which is nonsensical. When $AN_f$= 0.90, $OS_f$
=-33%, and when $AN_f$= 0.95, $OS_f$= - 50%. This indicates that in a sample containing some
mixture of OS, AS, and AN, the total sulfate would need to be 50% OS and 50% AS (at $AN_f$=
0.95) to give a non-negative $OS_f$. Thus for laboratory data, the Chen method should not be used
on mixtures containing $AN_f$> 0.50.

The effect of OA internally mixed with AS on the sulfate fragmentation pattern was also

explored with toluene, alkanol, and monoterpene SOA. For the alkanol SOA experiments we
found that the presence of even a small coating of alkanol SOA (which is thought to be liquid
(Liu et al., 2019)) shifts the normalized AS [1,1] point to ~[1.08,1.08], but increases in the
fraction of OA ($OA_f$) from 0.1 to 0.3 lead to no further changes in $nfH_ySO_x^+$ (Fig. 1B). This
means that for a sample containing a mixture of AS and alkanol SOA, the calculated $OS_f$ would
be -15% (Chen). In contrast, toluene SOA, which spans $0 < OA_f < 0.5$, shows no clear change in
the $nfH_ySO_x^+$ ions, indicating that $OA_f$ would not bias the Chen method for this example. The
monoterpene SOA, from two different experimental datasets using different AMSs, show more
varied results than the previous two studies. Overall, in the $OA_f$ range 0-0.50, the 2015
monoterpene data shows a consistent and constant 10-20% increase in $nfH_ySO_x^+$ compared to the
pure AS calibration point (similarly to the alkanol SOA). However, when $OA_f$ is in the range of
$0.50 < OA_f < 0.70$, 30-40% increases are observed. This result is only applicable to a few of the
experiments (see Fig. S6), potentially due to very high SOA loadings (up to 300 µg m$^{-3}$). These
high OA concentrations could potentially lead to a change of the particle phase due to
condensation of more volatile and liquid species, potentially altering the interactions of the
particles and the vaporizer surfaces.
These experiments collectively suggest that a "pure" AS calibration point of [1.15,1.15]
may be more appropriate when applying the Chen et al. method to some mixed aerosol at typical
OA concentrations observed in the atmosphere.
Chen et al. briefly discussed the potential impact of acidity on their OS quantification
method. This is explored here with pure sulfuric acid lab calibrations (Fig. 1C). Pure sulfuric
acid shows a large deviation from the pure AS triangle point (similar to increasing $AN_f$), nearly





doubling the values for the $nfH_ySO_x^+$ ions. This implies that a particle containing sulfuric acid
would produce a strong negative bias on the estimate of OS by the Chen method.

**3.2 Physical Interpretation of the Sulfate Fragmentation Trends**

It is useful to provide a physical interpretation of the trends that are likely driving the
observed sulfate fragmentation changes, based on the physicochemical details of the AMS
detection and those of the particles being sampled. In Fig. 2A, a simplified diagram of the AMS
detection process is shown, highlighting important details that are thought to give rise to the
observed trends.
Ambient particles containing AS, OS, and other species are sampled into the AMS
through a focusing lens. Following a series of differential pumping steps through the instrument,
the particles impact on a porous tungsten standard vaporizer. The time spent under vacuum from
sampling to detection is of the order of 15 ms. A fraction of the more viscous particles may
bounce from the vaporizer without detection. Non-refractory species in the particles that stick to
the vaporizer (such as OS and AS) are heated by heat transfer from the vaporizer surface. Some
species may evaporate in the form in which they are present in the particle, while others may
thermally decompose to other species, which then evaporate. For example, ammonium sulfate
may evaporate to $H_2SO_4(g)$ and $NH_3(g)$, but it may also thermally decompose to $SO_2(g)$, $SO_3(g)$
and $H_2O(g)$ (Hu et al., 2017b). Finally, these gaseous thermal decomposition products undergo
electron ionization to become positively charged species. Since the electrons used in EI have far
more energy (70 eV) than typical bonds in a molecule (~6 eV for S=O), the initial ions may
fragment into smaller ions if the ionization process results in absorption of > 6 eV of internal



energy by the molecule, beyond the ionization energy (Lambert, 1998). Some of the evaporated
$H_2SO_4(g)$ may remain as $H_2SO_4^+$ after ionization, or it may fragment to $HSO_3^+$ or $SO_x^+$ ions.
$SO_2(g)$ can only produce $SO_x^+$ ions. Thus the mixture of fragments observed will retain some
memory of the species that evaporated from the particles. If the mixture of evaporating species is
influenced by the particle composition (e.g. pH, AN, OA, or $OS_f$) then it may be possible to
calibrate the observed relationship to estimate a particle intensive chemical property.

Fig. 2A also shows a schematic close-up of the SV surface, which is the main point in the

instrument that controls ammonium sulfate fragmentation. In this diagram, we show a
non-smooth surface with pores, consistent with the fabrication of the vaporizer by sintering 50
μm tungsten spheres. The interaction of a particle with this porous surface is dependent on the
particle phase / viscosity. The red particles represent rigid (more solid-like) particles. These rigid
particles can simply bounce off of the vaporizer, leading to no detection. AS-dominated particles
are likely to be rigid (due to the solid phase of pure AS), thus increasing bounce and lowering the
AMS CE (Matthew et al., 2008; Middlebrook et al., 2012). AS particles can also become trapped
in the porous surface. When trapped, they are heated by conduction from the vaporizer surface
and by radiation from surrounding surfaces. They reach higher temperatures that lead to more
thermal decomposition, and a lower $H_2SO_4(g)/SO_x(g)$ ratio. In addition, molecules that evaporate
as $H_2SO_4(g)$ from these trapped particles are likely to collide with tungsten surfaces on their way
out to the ionization region, leading to additional thermal decomposition (Hu et al., 2017b) and
further reducing the $H_2SO_4(g)/SO_x(g)$ ratio for the gases reaching the EI region, and thus the
$H_ySO_x^+/SO_x^+$ ion ratio.



The second case (blue particle) represents the situation where the particle is less
rigid/viscous or liquid. Acidic sulfate particles (with a lower fraction of the sulfate ions
neutralized by $NH_4^+$), particles with high $AN_f$, or particles coated with a large water or liquid
organic layer are more likely to deform upon impact and not bounce. This leads to an increased
CE (Matthew et al., 2008; Middlebrook et al., 2012; Hu et al., 2017a). There are several effects
that will lead to a higher $H_2SO_4(g)/SO_x(g)$ ratio reaching the ionization region in this situation:
(a) evaporated $H_2SO_4(g)$ from particles that impact the front of the vaporizer and do not bounce
can now escape without further collisions with the tungsten surface; (b) the increased surface
area from impact deformation and the lower viscosity allow more $H_2SO_4(g)$ molecules to escape
the particle before those molecules are heated to temperatures that would lead to thermal
decomposition.
In Fig. 2B, we show a conceptual model of the impact of these phenomena on the Chen
triangle. For very acidic sulfate (approx. pH < 0), the liquid character of the particles leads to
less bounce in the vaporizer. It also leads to faster evaporation, which reduces the internal
temperature for the particles and that of the evaporated molecules, leading to less fragmentation.
In this regime $OS_f$ cannot be estimated, but pH may be, as long as it can be assumed (or shown
by additional measurements from the AMS or other instruments) that $OS_f$ and non-volatile
cations are small.  As an air mass becomes more neutralized by $NH_4^+$, the particles become less
acidic and more rigid/viscous, leading to more thermal decomposition of the evaporated species,
and the fragmentation of ammonium sulfate occurs at the upper vertex of the triangle. In this
regime methods such as Chen et al. (2019) may be applicable to estimate $OS_f$, as long as there
are no other effects that interfere with the sulfate fragments detected (such as substantial





non-volatile cations or variations in possible OA effects). As more ammonia is added to an
airmass, the acidity of the particles decreases and the higher pH favors the partitioning of
$HNO_3(g)$ to the particle phase, forming ammonium nitrate. If $AN_f$ becomes high enough ( $> 0.3$),
the particles again become less rigid/viscous and the fragmentation shifts again outside the Chen
triangle for the same reasons discussed for the acidic particles. Finally, Fig. 2C shows the
differences in the detection process and the fragments produced in the AMS for OS and/or
$AS/H_2SO_4$.

*3.3 Evaluation of the Chen Method with Aircraft Field studies*

The results of applying the Chen et al. (2019) method to five aircraft campaigns are

shown in Fig. 1D. The effect of internally mixed ammonium nitrate (AN) was explored in Fig.
1A and Sect. 3.1 (for laboratory studies). Here we explore the effect for field data from
KORUS-AQ (near Seoul, South Korea) where AN was often a major aerosol component;
average $AN_f \sim 0.18$). As discussed in Sect. 3.1, as the percent of AN in laboratory mixtures of
AS/AN increases, so do the $nfH_ySO_x^+$, ions. The same effect is observed for the KORUS-AQ
campaign, although the departure from the AS vertex is observed at substantially lower AN
fractions for the field data ($AN_f \sim 0.30$). When field data is affected by AN, the Chen method
might be applicable for situations with $AN_f < 0.30$. At higher fractions, a correction could
potentially be developed, but with increased resulting uncertainty.

In Fig. 1D, average values for each campaign in less acidic (pH > 0) and lower $AN_f$( <

0.3) conditions are shown. In the absence of acidity, OS, or $AN_f$ effects, it is expected that the
data would fall on top of the [1,1] pure AS point in the 1D triangle plot, but this is not observed.





This shift suggests that there are other factors (such as the presence of organics) that affect the
location of the pure AS point. In addition, the average values for the different campaigns vary
substantially, so it is unlikely that a "corrected" pure AS point can be used for all campaign
and/or lab data.
To further look into the potential effect of acidity, we consider the ATom campaigns in
Fig. 1C. ATom focused on remote oceanic air, with very low $AN_f$ (< 0.008). This is expected as
AN is semivolatile (DeCarlo et al., 2008; Hennigan et al., 2008; Nault et al., 2018) and for the
very low pH conditions during ATom (~ -1 to 1, average of -0.6), most of the nitrate will be in
the form of $HNO_3(g)$ (Guo et al., 2016). The PALMS instrument independently reports $OS_f$ ~ 0.3
- 0.7% for ATom (depending on the pH). The results for ATom span the range between pure AS
and pure $H_2SO_4$, following a monotonic trend as acidity increases, consistent with the laboratory
results and the results from the WINTER campaign in Chen et al. (2019). We hypothesize that
high acidity is leading to the observed departure from the Chen triangle. Hence, the ATom results
suggest that all of the sulfate sampled is inorganic and if the Chen method is applied $OS_f$ = -26%
to +4%.  Thus the Chen method is insufficient to describe the trends observed for very acidic
aerosols, until pH increases to ~ 0 (where the ATom data starts to converge onto the pure AS data
point). For campaigns containing particles of pH > 0, the Chen method might be applicable.
To further illustrate that the ATom and KORUS-AQ campaigns are representative of the
range of airmasses in the troposphere, Fig. 1D shows results for two additional campaigns that
focused on the continental US. SEAC[4]RS and WINTER represent chemical regimes that are not
extremely acidic (average pH SEAC[4]RS ~ -0.2, WINTER pH ~ 1.2). SEAC[4]RS had low $AN_f$ ( ~
0.04), while WINTER had high $AN_f$ (~ 0.25). It is observed that every single campaign average



falls outside of the triangle (for the full campaign and non-acidic, low $AN_f$ averages), indicating
that the Chen et al. method, as proposed, breaks down for many regions of the atmosphere.
Average $AN_f$, $OA_f$, and pH values for different campaigns are shown in table S1.

**3.4 Specification of aerosol chemical regimes for feasibility of $OS_f$ quantification**
In Fig. 3A, we introduce a plot of $AN_f$ vs. pH that can be used to evaluate the
applicability of the $OS_f$ methods to different datasets. Data for five different campaigns (those
with AS calibrations, labelled "C" in Table 1) are shown, along with the campaign averages.
Regime I ("highly acidic, low AN") occupies the bottom left quadrant, where $AN_f < 0.3$ and pH
$< 0$. Campaigns sampling the more remote atmosphere (e.g. ATom-1, 89% of datapoints;
ATom-2, 80%), and a fraction of the data from continental campaigns (SEAC[4]RS, 13% ; DC3 ,
40%) fall in this regime. For remote regions, emissions (such as $NH_3$ and $NO_x$) are generally low.
Remote oceanic regions are relatively isolated from the major continental ammonia sources
(Paulot et al., 2015). Therefore, less ammonia is available to balance the hydronium ions from
$H_2SO_4$, leading to high acidity (Quinn et al., 1988; Keene, 2002; Nault et al., 2020). Highly
acidic aerosols and lack of $NH_3$ shift $HNO_3$ to the gas phase, so low $AN_f$ is observed. In contrast,
for sampling in polluted source regions with strong $HNO_3$ formation and substantial $NH_3$
emissions, a much smaller fraction of the data falls in this regime (e.g. only 4% for
KORUS-AQ). In Sect. 3.5 we discuss the potential to estimate pH from AMS data in regime I.
Regime II (lower right) involves less acidic conditions (pH > 0) and lower $AN_f$ (< 0.3). In
this region sulfate fragmentation in the AMS is not strongly impacted by either $AN_f$ or acidity. In
principle, in this regime the recently proposed sulfate deconvolution methods can be used. The





geographical regions studied in Chen et al. (2019) and Song et al. (2019) generally fall in this
regime, and this explains the lack of large negative $OS_f$ values in those studies, in contrast to our
observations for other regions. About half of our campaign data is located in this regime, more
so for the continental campaigns and much less for the remote campaigns. Specifically, 65% of
KORUS-AQ, 60% of DC3, 87% of SEAC$^4$RS, 11% of ATom-1 and 20% of ATom-2 fall in this
regime. We have applied the 1D version of the Chen method to each field campaign after
filtering it by the $AN_f$ and pH constraints for regime II. $OS_f$ is nominally slightly greater than
zero for ATom-1 ($OS_f \sim$ 3%, greater than the 0.3% estimate in regime II from PALMS (for
ATom-1 and ATom-2, estimated by only considering the sulfate moiety from the IEPOX or
glycolic acid sulfate (GAS) OS, neither of which was detected in the supermicron aerosol (Froyd
et al., 2009, 2019; Liao et al., 2015)) but still small, (see Fig. S7) much less than zero for
ATom-2 ($OS_f \sim$ -23%) and KORUS-AQ ($OS_f \sim$ -26%). This shows that even when pH and AN
are not major influences on the sulfate fragmentation, estimating OS with sulfate ions may be
susceptible to inaccuracies in AS calibrations, noise present in the ambient data, or other factors.
We also show results from applying the Song et al. (2019) method in regime II (which is
based on similar principles to the Chen method) in Fig. S8. Similarly to the Chen method, we see
that most $OS_f$ values are predicted to be less than zero. For the entire atmosphere, shown in Fig.
S9, the distribution for% OS looks similar to Fig. S8.
Regime III is characterized by high $AN_f$ (> 0.3) and lower acidity (pH > 0). This
chemical regime primarily exists in polluted continental regions near large source regions such
as megacities and agricultural regions, as high $NO_x$ and $NH_3$ emissions can lead to increased
particulate AN and an increase in aerosol pH (Pye et al., 2019). In this regime, there are strong


variations in the AMS sulfate fragments that are driven by $AN_f$. $OS_f$ cannot be estimated with the
AMS sulfate fragmentation methods proposed so far, unless they are further modified to account
for the $AN_f$ effect. ~ 31% of KORUS-AQ data falls in this regime, but almost none of the data
from the rural / remote campaigns falls in this region, as AN typically evaporates as the air is
diluted during advection away from polluted regions (DeCarlo et al., 2008).

Finally, regime IV in the top left quadrant has high AN ($AN_f > 0.3$) and high acidity (pH

< 0). This chemical regime is unlikely to be observed in the real atmosphere, and indeed there
are very few points in that region for our campaigns. Sulfate is ubiquitous (Zhang et al., 2007b;
Hodzic et al., 2020), and nitrate is not thermodynamically stable in the aerosol phase together
with acidic sulfate for pH < 0 (Guo et. al., 2016). For all campaigns we observe ~ 0% of points
occupying this regime. Very unusual datapoints can be observed when ammonium
nitrate-containing particles are externally mixed with acidic sulfate containing particles in an
airmass.

Since the field studies analyzed here targeted large regions but did not sample many

others, it is of interest to evaluate the fraction of the troposphere located in each one of the
chemical regimes. The results of the GEOS-Chem v12 model are used for this purpose in Fig. 3B
and shown as a global map in Fig. 4 and Fig. S10. ~ 67% of the model troposphere exists in
regime I (pH < 0). In addition, ~ 33% of the global troposphere exists in regime II where it may
be feasible to estimate $OS_f$ from AMS fragments. Less than 1% of the modeled atmosphere
exists in regime III (upper right quadrant) where ammonium nitrate strongly influences sulfate
fragmentation, consistent with the relatively small very polluted geographical regions with very
large $AN_f$. Finally, none of the data fell in regime IV, consistent with aerosols being assumed to





be internally mixed in GEOS-Chem. At the surface during December, January, and February
(DJF) (Fig. 4A), most of the remote oceans fall in regime I (pH < 0 and $AN_f$ < 0.3), while regime
II (pH > 0 and $AN_f$ < 0.3) is dominant over continental regions. At the surface in June, July, and
August (JJA) (Fig. 4C), most of the globe is in regime II. Very little of the data falls in regime
III, except parts of Asia, regardless of season. A similar pattern is observed in the free
troposphere (Fig. 4B and 4D) with some geographical differences. Regime III (pH > 0 and $AN_f$ >
0.3), which represents pollution hotspots, is observed in a large region in Asia during the
Summer months, whereas the Winter months are dominated by regime I (low pH). The Summer
months in the free troposphere are also mostly in regime II, especially over continental regions.
Due to averaging of an entire year, as well as the limited spatial resolution of the GEOS-Chem
model, locations and periods of high $AN_f$ hotspots are not as prominent in these results, even
when the data is divided by season.

**3.5 Potential pH estimation from AMS measurements**

**3.5.1 Estimation of pH from AMS sulfate fragments**

In Sect. 3.4, we introduced chemical regime I with low pH and low $AN_f$. In this regime,

which encompasses about half of the campaign data and ⅔ of the modeled global troposphere,
PALMS data shows that the overwhelming majority of the sulfate is inorganic, with $OS_f$
contributing ~ 0.7% to total sulfate by mass during ATom-1 and ATom-2 when pH < 0 (see Fig.
S7, in regime I). This removes sulfate fragmentation changes caused by AN and sulfate type (OS
vs. AS), indicating that sulfate fragmentation is almost exclusively controlled by the acidity of
the aerosol. Fig. 1C shows that $fH_2SO_4^+$ and $fHSO_3^+$, i.e. the amount of sulfate fragments





retaining one or two hydrogens ($H_2SO_4^+$ and $HSO_3^+$) relative to the sulfate fragments without a
hydrogen atom ($SO_3^+$, $SO_2^+$, $SO^+$) increases as pH decreases.
In Fig. 5 we show the relationship between $H_ySO_x^+/SO_x^+$ and aerosol pH. As the
relationship is noisy for individual data points, we show the results for 5% quantiles of the data.
$H_ySO_x^+/SO_x^+$ appears to show a proportional relationship with decreasing pH for the ATom
campaigns, for which much of the data is in regime I. The KORUS-AQ data, of which very little
falls in the regime I, does not show a relationship between these variables, as expected. A fitted
equation to the ATom relationship may allow the real-time estimation of pH for different air
masses for campaigns in regime I as:

$$pH = -1.3\,(\pm\,0.06) + 6.0(\pm\,1.2) \times e^{-1.3(\pm0.18)\times\frac{H_ySO_x^+}{SO_x^+}} \qquad \text{Eq. 13}$$


As shown in the histogram in Fig. 5B, this relationship is applicable to a substantial fraction of
ambient observations. This estimation equation likely needs to be calibrated for each instrument
(e.g. by sampling sulfate particles with different acidities), since the sulfate fragmentation does
vary with instrument (Chen et al., 2019) (and potentially also in time for a given instrument).
Although an estimation equation that apparently works for only one unit of pH may seem
of limited value, two caveats apply: first, it is of high value to know that pH < 0 for a certain air
mass (as opposed to e.g. pH = 2 or 3 that are frequently encountered). Second, the range of pH
below 0 is limited here due to not considering the activity coefficient. If that coefficient was
included, the predicted pH range in this regime would be ~ -4 to 0.

**3.5.2 Estimation of pH from Ammonium Balance**





Ammonium balance ($NH_{4\_bal}$) (Eq. (7)) is often used as a qualitative indicator of acidity.
(Zhang et al., 2007a) showed that pH under constant temperature and RH was well correlated
with ammonium balance, but much more scatter was observed when the instantaneous T and RH
were used. Several studies have argued that ammonium balance cannot be used to estimate
ambient pH (e.g., (Guo et al., 2015, 2016; Hennigan et al., 2015; Weber et al., 2016); however,
those studies were all performed at continental ground sites that were in the less-acidic chemical
regimes (II and III), and where daily temperature and humidity changes were strong. As shown
in Fig. 6, $NH_{4\_bal}$ and pH for the aircraft studies show a strong and consistent relationship in
regime I (pH < 0), providing another potential method for estimating pH. As ammonium balance
increases, so does pH across the six campaigns studied. These data are generally outside of the
continental boundary layer, where temperature and RH change less in a diurnal cycle, reducing
the impact of those changes on pH. For data in regimes II-III (pH > 0) some proportionality of
pH and $NH_{4\_bal}$ is still observed on average, but with more dispersion across campaigns. Given
the similarity of the results for regime I, the fitting equation of pH vs. ammonium balance may
be used to provide a near real-time estimate of pH (for $NH_{4\_bal}$ < 0.65).

$$pH = -1.1(\pm0.031) + 1.7(\pm0.089) * NH_{4\_bal} \qquad \text{Eq. 14}$$


As shown in the histogram in Fig 6B-6D, this relationship is also applicable to a
substantial fraction of ambient regions. This estimation equation should be tested with other
studies. An advantage of this relationship (vs. the one based on $H_ySO_x^+/SO_x^+$) is that it is likely to
be less instrument-dependent, as long as careful calibrations of $RIE_{NH4}$ and $RIE_{SO4}$ have been
performed. Conditions where non-volatile cations (e.g. $Na^+$, $K^+$, $Ca^{2+}$) are important for





submicron particles could lead to deviations from this relationship (Guo et al., 2020). However,
such conditions are infrequent in remote air (Nault et al., 2020) and can be diagnosed by
concurrent supermicron or filter measurements.

### 665 3.5.3 Application of pH estimation methods to ambient data

As discussed above, ammonium balance and $H_ySO_x^+/SO_x^+$ are two measurements that
may be used to estimate aerosol acidity in parts of the atmosphere. In Fig. 7 these two methods
are applied to one flight during ATom-1 and an $SO_2$ plume sampled during WINTER. In Fig. 7A,
both $H_ySO_x^+/SO_x^+$ and $NH_{4\_bal}$ follow the trend for E-AIM calculated pH during most periods
when pH < 0, even at one minute time resolution.
As expected from Fig. 6, $NH_{4\_bal}$ is a less noisy, more robust metric for estimating pH at
one minute time resolution. Unlike $H_ySO_x^+/SO_x^+$, $NH_{4\_bal}$ appears to be able to capture basic pH
trends at the full range of pH values observed during this flight in ATom-1. $NH_{4\_bal}$ also matches
the E-AIM predicted pH well for the WINTER power plant plume. For RF01 in ATom-1
(WINTER), $NH_{4\_bal}$ predicted pH has an $R^2 \sim 0.6$ (0.9) for pH<0 (Fig. 7C-D).This shows that in
the remote atmosphere (like in ATom) or in an $SO_2$ plume, $NH_{4\_bal}$ has the potential to allow fast
estimation of pH, even under relatively low sulfate concentrations, albeit not perfectly. More
scatter is observed for the estimate based on $H_ySO_x^+/SO_x^+$, indicating that longer averages are
needed for this method. The error is typically within +/- 0.5 pH units, which is thought to be the
accuracy of thermodynamic pH estimation models.

3.6 Possibility of Estimating CE from Sulfate Fragmentation





From the previous discussion it is clear that sulfate fragmentation changes due to some of

the same factors (acidity, $AN_f$) that influence ambient AMS CE. It is of interest to explore
whether a quantitative estimate of ambient particle CE could be derived from the measured
sulfate fragments, at least under some conditions, as it could provide a complementary
characterization to the CE estimates from the Middlebrook et al. (2012) parameterization. In Fig.
8 we show the CE estimated from Middlebrook et al. (2012) vs. $H_ySO_x^+/SO_x^+$ for ATom and
KORUS-AQ. CE does show some relationship with $H_ySO_x^+/SO_x^+$, with most sensitivity around
CE ~ 0.8-0.9. A substantial level of noise is observed on the high-time resolution data, and the
trend varies between the two campaigns (where variations in CE are controlled by two different
effects, acidity vs $AN_f$). Further research would be necessary to evaluate whether this method
could be used to estimate CE.

**Conclusions**

The presence of organosulfates in particles is a topic of much recent interest, but there is

a lack of online methods to quantify them. Two methods have been proposed to use widely
available AMS data to quantify $OS_f$ (Chen et al., 2019; Song et al., 2019). However, these
methods have only been applied to ground continental datasets, to our knowledge. We show
using both laboratory and field data that both high acidity (chemical regime I in this work) and
high $AN_f$ (regime III) result in major changes in sulfate fragmentation, which often produce
nonsensical results for the $OS_f$ methods. Regime I accounts for ~ ⅔ of the global troposphere ,
while regime III can be important in polluted regions (e.g. Seoul region), and thus it is critical to
avoid applying the proposed $OS_f$ estimation methods in these regimes. In regime II, with lower


acidity and lower nitrate (pH > 0, $AN_f$ < 0.3) $OS_f$ estimation methods may be applicable, if no
other effects (e.g. significant non-volatile cations or variations in OA effects) confound the
sulfate fragmentation. For reasons not fully understood, fragmentation of the sulfate ions in the
lab vs. ambient data differ at times.
We investigated two different methods to estimate pH in real-time in regime I (pH < 0
and $AN_f$ < 0.3), based on the AMS $H_ySO_x^+/SO_x^+$ fragment ratio and the ammonium balance,
respectively, without the need to run a thermodynamic model, and without the need for gas-phase
$NH_3$ or $HNO_3$ measurements. Low $OS_f$ and non-volatile cations need to be assumed or confirmed
from AMS and other measurements. The ammonium balance method shows better performance.
These *in-situ* and direct pH estimation methods should be applicable in the remote atmosphere
(oceanic regions, and often the continental free troposphere when not recently impacted by
surface sources). Both the $OS_f$ and pH estimations require careful instrument calibration for a
given campaign, and the methods based on sulfate fragments are expected to be
instrument-dependent, including for the same instrument in time when filaments or the vaporizer
are replaced, or when the instrument is re-tuned. Both methods should be further evaluated with
data from other studies.
We propose a conceptual model to explain the observed sulfate fragmentation changes
with changing particle chemical composition. As particles become more acidic or higher in AN,
a higher fraction of $H_2SO_4(g)$ can reach the ionization region, leading to changes in the observed
ion population. Since AMS CE is thought to be controlled by the same effects, we explore
whether it can be estimated from the observed sulfate fragmentation, and find that while changes
in $H_ySO_x^+/SO_x^+$ do correlate to changes in CE, the relationship is not the same across different



campaigns. Further investigation of this relationship, especially when direct CE measurements
are available via internal AMS light scattering, would be of interest.

**Acknowledgements**
This work was supported by NASA grants NNX15AH33A & 80NSSC19K0124, and a CIRES
IRP project. We thank the members of the Jimenez group, the AMS users community, Weiwei
Hu, Amber Ortega, and Patrick Hayes for help with data acquisition during SEAC$^4$RS and DC-3;
Jason St. Clair, Alex Teng, Michelle Kim, John Crounse, and Paul Wennberg for providing
CIT-CIMS $HNO_3$ data; Joel Thornton, Felipe Lopez-Hilfiker, and Ben Lee for providing
UW-CIMS $HNO_3$ data during WINTER; Karl Froyd, Gregory P. Schill, and Daniel Murphy for
providing PALMS organosulfate data for ATom campaigns; and Glenn Diskin for providing
DLH $H_2O$ data.

**Data Availability**
DC3 data available at DOI: 10.5067/Aircraft/DC3/DC8/Aerosol-TraceGas, last accessed on 9
September, 2018. SEAC$^4$RS data available at
http://doi.org/10.5067/Aircraft/SEAC4RS/Aerosol-TraceGas-Cloud, last accessed on 27 April,
2020. WINTER data available at  https://data.eol.ucar.edu/master_lists/generated/winter/, last
accessed 27 April 2020. KORUS-AQ data available at DOI:
10.5067/Suborbital/KORUSAQ/DATA01, last accessed 14 June, 2018. ATom-1 and ATom-2
data available at https://doi.org/10.3334/ORNLDAAC/158, last accessed 27 April 2020.



**Tables**

*Table 1. Summary of the campaigns used in this study. See SI Fig. S1 for flight paths. Reference label refers to the type of data used for each campaign throughout this paper, depending on the quality and completeness of the data, for the purposes of a specific analysis. A : ammonium balance, f: $SO_4$ campaign-averaged fragments, F: $SO_4$ campaign-average and time-resolved fragments , and C: pure AS calibration data reliable and used.*

| Campaign | Location | Season/Year | References | Reference Label |
|---|---|---|---|---|
| DC3: Deep Convective Clouds and Chemistry | Mid-Latitude Continental United States | May-June 2012 | Barth et al. , 2015 | A |
| SEAC[4]RS: Studies of Emissions and Atmospheric Composition, Clouds and Climate Coupling by Regional Surveys | Continental United States | Summer 2013 | Wagner et al., 2015; Toon et al., 2016 | A, f, C |
| WINTER: Wintertime Investigation of Transport, Emissions, and Reactivity | Eastern United States, continental and marine | Winter 2015 | Schroder et al., 2018; Shah et al., 2018 | A, f, C |
| KORUS-AQ: Korean United States Air Quality | South Korean Peninsula and Yellow Sea | Spring 2016 | Nault et al., 2018 | A, F, C |
| ATom-1: Atmospheric Tomography Mission 1 | Remote Pacific and Atlantic Basins | Boreal Summer 2016 | Hodzic et al., 2020; Brock et al., 2019; Hodshire et al., 2019 | A, F, C |
| ATom-2: Atmospheric Tomography Mission 2 | Remote Pacific and Atlantic Basins | Boreal Winter 2017 | Hodzic et al., 2020 | A, F, C |










Fig. 1. Laboratory and field data for sulfate fragmentation shown in the triangle diagram
proposed by Chen et al. (2019). (A) Data split into 10 quantiles of $AN_f$ value for the full
KORUS-AQ campaign, as well as for different laboratory internal mixtures of AS and AN . (B)
Data from two chamber experiments, split into 5 quantiles of $OA_f$. Data with very high OA
(>100 µg m$^{-3}$) are shown as grey triangles. Two separate datasets of monoterpene SOA chamber
experiments are labelled as "2014" and "2015". (C) Data split into 10 quantiles by pH for
ATom-1 and 2, colored by pH from E-AIM. (D) Averages for 5 aircraft campaigns for the full
campaign and a subset of each campaign where pH<0 and $AN_f$<0.3.



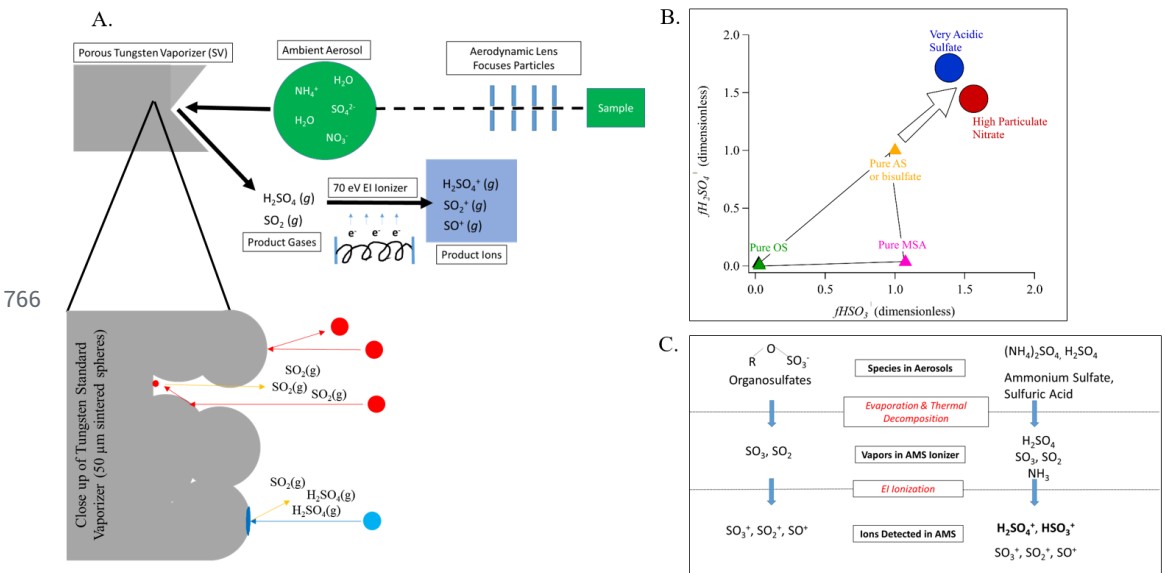

Fig. 2. (A) Simplified schematic of the AMS detection process, including a close up of the tungsten standard vaporizer surface and the different species produced by AS and OS. (B) Conceptual model of the position of particles of different compositions in the Chen et al. (2019) triangle plot. As particles become more acidic or higher in particulate nitrate, the ratio of the AMS hydrogenated to total sulfate fragments increases. When sulfate is present as AS (or mixtures of AS and ammonium bisulfate), the sulfate fragmentation is mainly impacted by OS vs. AS vs. MSA relative concentrations inside the Chen triangle. (C) Schematic of the transformations during the AMS detection process for OS and AS.



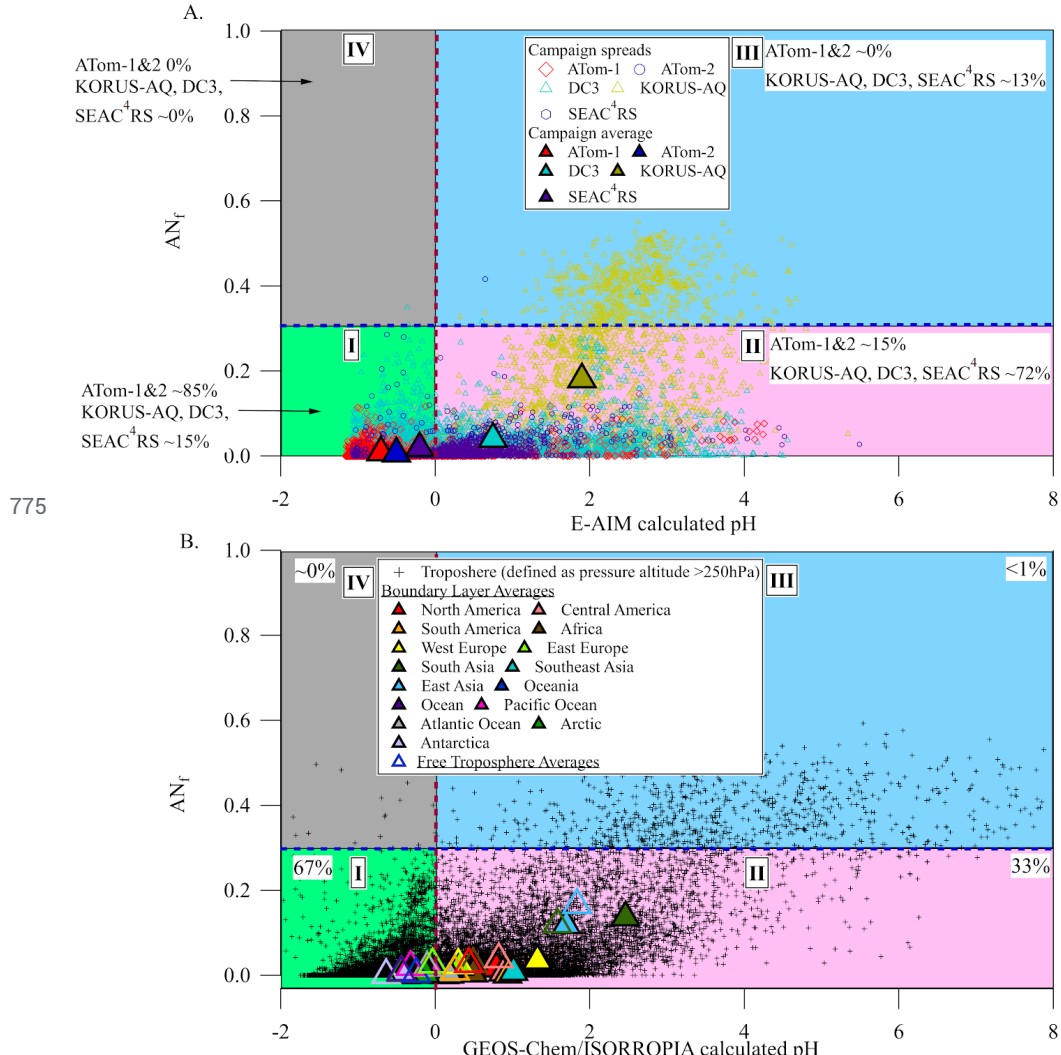

Fig. 3. (A) location of the aircraft campaign 1-minute data points on the chemical regimes
defined in this paper ($AN_f$, from AMS measurements) vs. E-AIM pH. SEAC[4]RS, WINTER, and
KORUS-AQ are averaged to one value, for brevity, but defined individually in Sect. 3.4. (B)
Location of global GEOS-Chem v12 results in the chemical-regimes diagram. Yearlong averages
shown as large triangles.



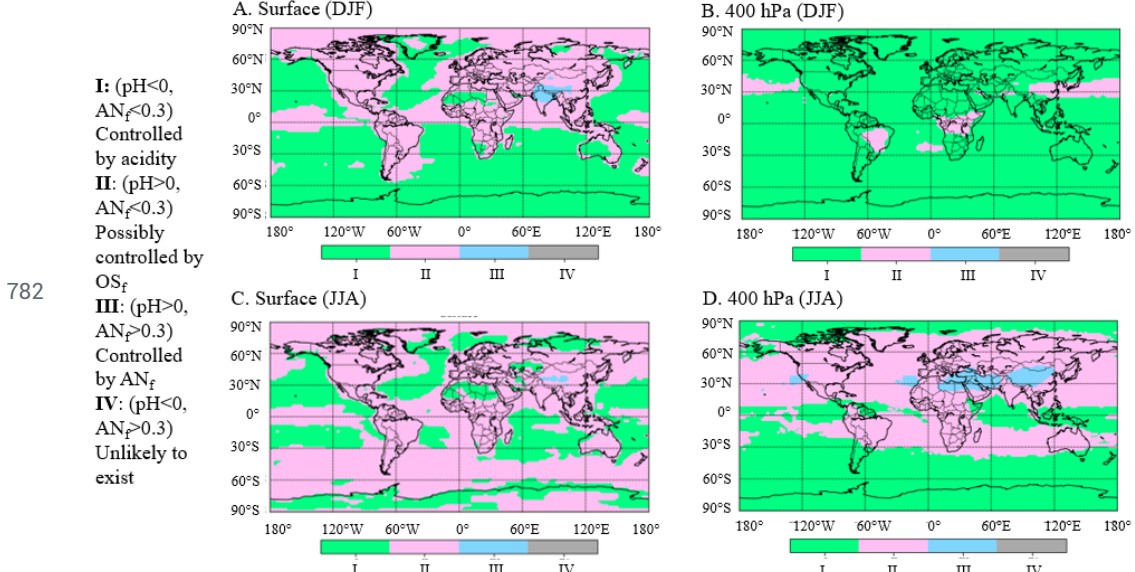


**I:** (pH<0, $AN_f$<0.3) Controlled by acidity
**II:** (pH>0, $AN_f$<0.3) Possibly controlled by $OS_f$
**III:** (pH>0, $AN_f$>0.3) Controlled by $AN_f$
**IV:** (pH<0, $AN_f$>0.3) Unlikely to exist

Fig. 4. Areas characterized by different chemical regimes according to results from
GEOS-Chem v12. (A) Surface for December, January, and February (DJF), (B) 400 hPa for DJF,
(C) Surface for June, July, and August (JJA), (D) 400 hPa for JJA. Roman numerals correspond
to regimes in Fig. 3.




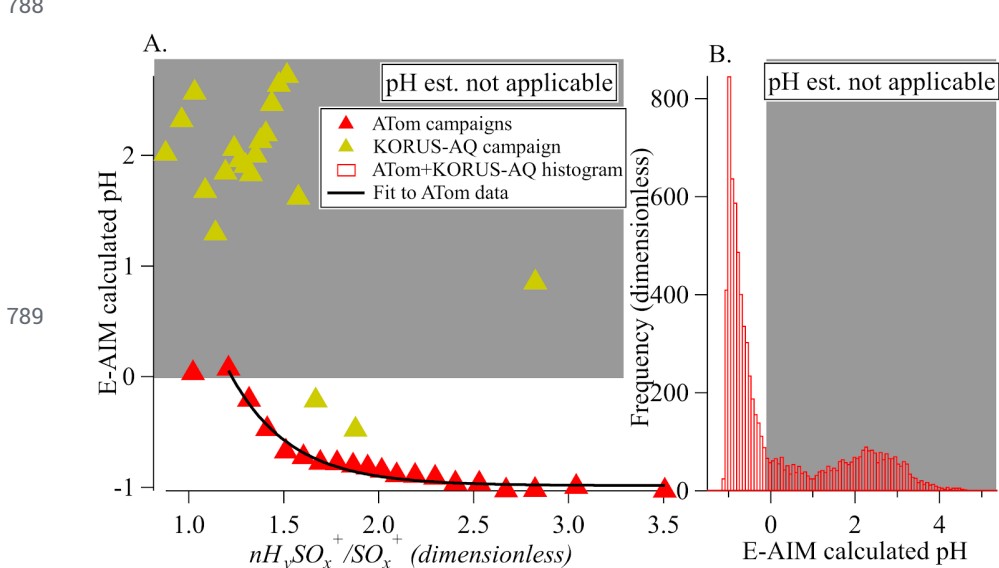


Fig. 5. (A) pH vs. sulfate fragmentation indicator ($H_ySO_x^+/SO_x^+$) for the ATom and KORUS-AQ campaigns, and binned by $nH_ySO_x^+/SO_x^+$. The black line is an exponential fit to ATom data (see text). (B) histogram of the calculated pH for the 1-minute datapoints from the ATom-1,2 and KORUS-AQ datasets. In both panels, the white (gray) area shows the regime where pH can (cannot) be estimated from the sulfate fragmentation.





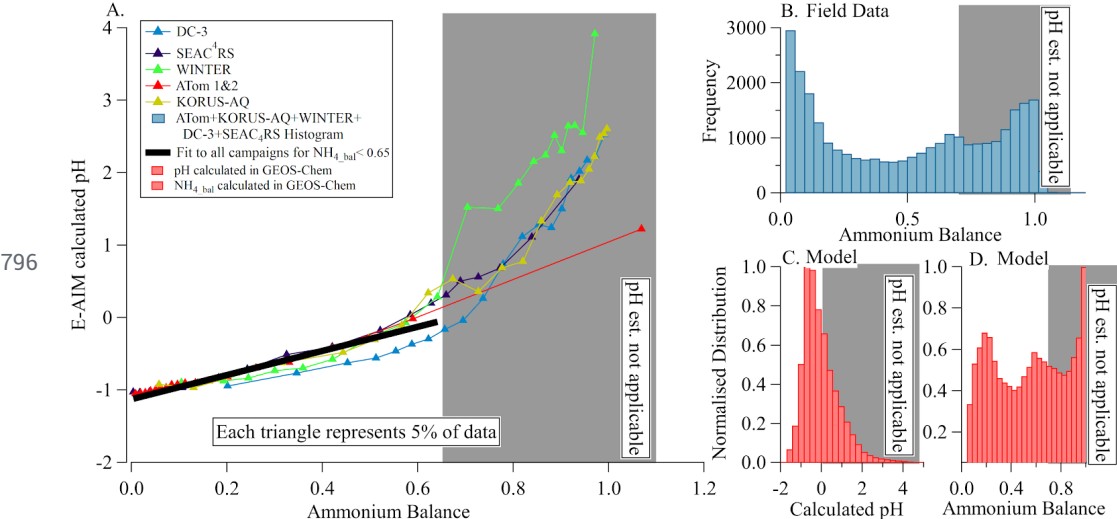

Fig 6. (A) calculated pH vs. ammonium balance for multiple campaigns. Quantiles of the data
are used to reduce the impact of noise. The black line is an orthogonal distance regression (ODR)
fit to the campaign data for values with $NH_{4\_Bal} < 0.65$. B) Histogram of measured ammonium
balance for the 6 campaigns. (C) and (D), pH and ammonium balance from GEOS-Chem (pH
calculated with ISORROPIA). In all panels the white (grey) areas encompass the data points for
which pH can (cannot) be estimated from the measured ammonium balance.







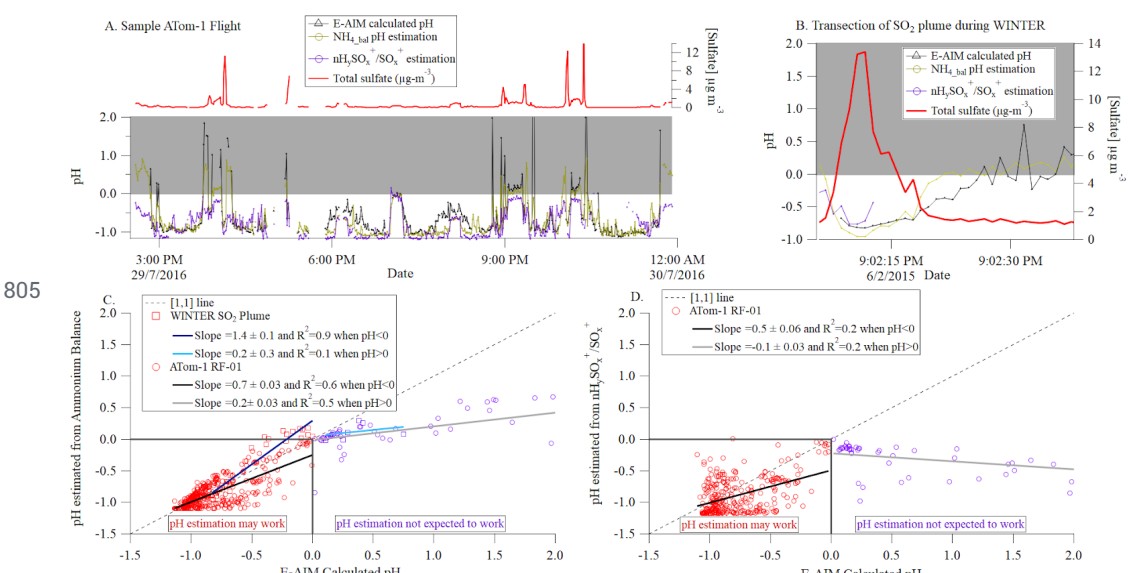

Fig. 7. (A) Time series of sulfate, pH calculated from E-AIM and estimated from $H_ySO_x^+/SO_x^+$,
and $NH_{4\_Bal}$ for one flight during ATom-1 (at 1 min. resolution, filtered to remove points where
sulfate was less than 3 times its detection limit). (B) Time series of sulfate and pH for a large
power plant plume sampled during WINTER. (C) Scatterplot of pH predicted from $NH_{4\_Bal}$ vs.
E-AIM pH for the data above. (D) Scatterplot of pH predicted from $H_ySO_x^+/SO_x^+$ vs. E-AIM pH
for the ATom flight.






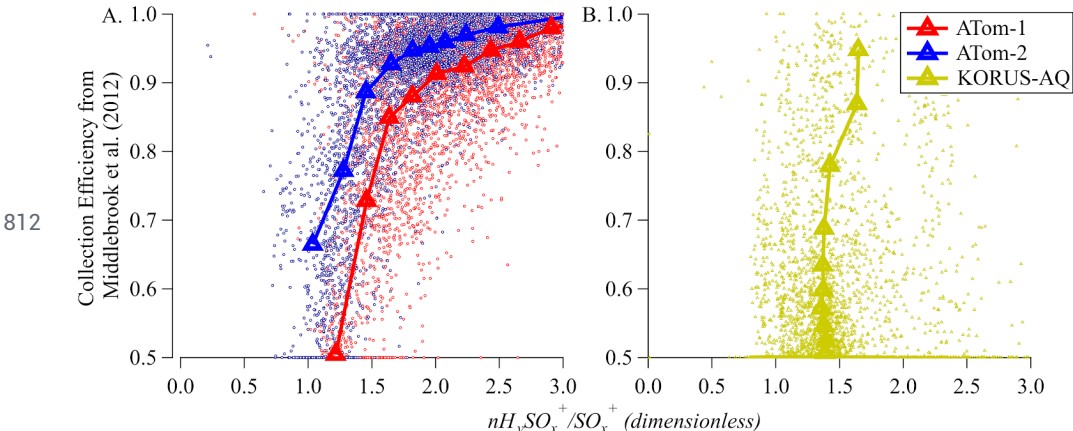

Fig. 8. (A) Collection efficiency parameterization vs. $H_ySO_x^+/SO_x^+$ for two ATom campaigns, and
(B) the KORUS-AQ campaign.






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
