# Peer review of "Aerosol pH Indicator and Organosulfate Detectability from Aerosol Mass Spectrometry"

_Atmospheric Measurement Techniques, 2020_

## Referee Comment (RC1) · Anonymous Referee #1 · 4 Nov 2020

Review for Aerosol pH Indicator and Organosulfate Detectability from Aerosol Mass Spectrometry Measurements

Summary:

This paper discusses the challenges of using AMS sulfate ion fragments to distinguish organic sulfates from inorganic sulfates and relates sulfate ion fragment ratios and ammonium balance to aerosol pH calculated from E-AIM. The authors apply previously published methods [Chen et al., 2019 and Song et al., 2019] to several past airborne AMS datasets to test the applicability of these methods for many aerosol compositions. They also determine sulfate ion fragment ratios for mixtures of ammonium sulfate (AS)

with inorganic and organic species in laboratory experiments. While these methods should be useful in some regions (pH>0, low ammonium nitrate - AN) based on the laboratory results presented here, the authors have determined that these methods often produce nonsensical values (negative OSf enrichment), suggesting that these methods cannot be used universally across all aerosol compositions. Specifically, high AN and high acidity change the way that sulfate interacts with the AMS vaporizer and prevents use of the Chen method in polluted (high AN) and remote marine regions (high acidity). However, even in the regions that should give good results (low AN, low acidity), the calculated organic sulfate fraction is often negative. The authors also show that sulfate ion fragments ratios correlate to E-AIM-calculated pH under certain conditions (pH

around, especially when describing figures. For example, Fig. 1 is complex with a lot of data (lab plus field campaigns). Parts of this figure are described for the first time in disjointed places: Sect 3.1, 3.3 (but not 3.2), and 3.5.1. To help connect the laboratory and field applications of the Chen method, I suggest that the description and discussion of Fig. 1 be presented in its entirety and Section 3.2, which contains explanations for how the AMS responds to various species, could be reduced, moved, or integrated into the discussion of Figure 1.

Specific Comments:

58. Can you/ did you apply these methods to ACSM data? It seems that assumptions about hydrocarbon contributions at unit mass resolution, especially at m/z 81 (HSO3+, C6H9+, C5H5O+), would further increase uncertainty in this quantification. Just an interesting thought, since the ACSM was brought up here.

76. Please check that these are the intended citations. I don't see organic sulfate fractions in Riva 2019a "Evaluating the performance of five different chemical ionization techniques for detecting gaseous oxygenated organic species," so perhaps that is not the correct reference.

91. Instead of writing "inorganic" in quotations, be please specific: that these are ions that do not contain carbon and are typically categorized as /associated with inorganic sulfate. However, organic sulfate can also produce these ions.

93. Decomposition or evaporation + ionization rather than ionization/decomposition? Section 3.2 and Fig 2C does a nice job of explicitly showing which species evaporate or thermally dissociate (and ionize) to form which ions, so perhaps use the same ideas and consistent language here and move parts or all of Fig 3 to the introduction.

Table 1. Be consistent. i.e. DC3: Summer 2012. ATom-1: Boreal Summer/ Austral Winter [sampling covered both northern and southern hemispheres at the same time but in different seasons, correct]?
266. Definition of HySOx: The distinction between its definition in Eq 5 (HySOx = H2SO4 + HSO3) and Eq 10 (HySOx = H2SO4 + HSO3+ SO3) is clear in the text, but it's annoying to have two definitions (and quantities) for the same variable in one paper. The distinction is lost when first introduced (total sulfate signal is defined on In 96, but an undefined HySOx is used on In 106).

352. What altitude is this? This reference to the DC-8 seems out of context. Does this paragraph only address marine amines at minimum altitudes during ATom only?

347-364. Consider moving this paragraph from the methods to discussion section

387. The thing that strikes me about Figure 1 is that even though "data are expected to lie inside the triangular region," few of the lab or field averages do! The AS lab cases that should show have fragment ratios of pure AS, do not, and this is explored. The field campaign averages also do not have fragment ratios consistent with calibration AS, and this is partially explored. Stating this discrepancy up front would improve readability of the discussion section.

395. I think this example is unnecessary. If I'm understanding correctly, even if 50:50 OS:AS gives a non-negative OSf in the ANf= 0.95 case, the method would indicate there is no OS which is also not correct. Another way to say this is that as the endpoint for AS with high AN moves away from pure AS due to the interference from AN, the method estimates OSf with increasing inaccuracy so that it would predict OSf= 0%, when OSf is really 50% in this extreme case. So, even though the result (OSf = 0) is a "reasonable" non-negative OSf, it is still completely incorrect.

399. Figure 2b does not effectively show the information in this paragraph since the points overlap on a small region of the plot. What is the ratio of OA/AS when OA > 100 ug/m3? All other experiments are in relative terms, so 100 ug/m3 does not have any context here. How does OA/AS ratio compare to that of the campaign averages? The regime II averages?
415. The hypothesis here is that you can change the relative HxSOy fragments due to different residence times on the vaporizer for OA-coated particles (i.e. less bounce, more thermal decomposition)? Update: I see this gets discussed more in Section 3.2, so perhaps it would be better to introduce these ideas within Section 3.1.

425. I suggest putting Section 3.2 somewhere else. Before 3.1? After 3.3? Figure in the SI with the key points in the discussion of Fig 1? Moving the generalize fragment discussion (Fig 3C) to the intro? Figure 1 includes both lab and field data, and it seems odd that the discussion of a single figure is broken up like this without any indication that the discussion will be picked up again. I got to the end of Section 3.1 wondering about Fig 1D and why all the campaign averages are out still of the "Chen triangle." It seems to me that Section 3.2 contains the analytical crux of why this method isn't universally applicable, so this is an important section. It just seems in a weird place. This discussion of the AMS response is applicable for both the lab and field data, so I think it makes sense to describe Figure 1 in it's entirety, and then, as a discussion, talk about the fragmentation for these different conditions (in context of both the lab experiments and the complex mixtures in the ambient data).

458. Based on this explanation, that particles with longer contact time with the vaporizer experience higher temperatures and thus more thermal decomposition, I would expect the sulfate ion fragment ratios to depend on the vaporizer temperature. Has sensitivity to vaporizer temperature been tested?

506. Define pH>0 and AN<0.3 as Regime II here in the text. Combine this paragraph with In 526-533.

533. This is a general comment: I'm struggling with the paper setting up the discussion that even though the proposed method does not work for high acidity or high AN as shown in the lab results, it will work for subsets of ambient data that meet the criteria (pH>0; AN

moving the AS fragment ratio away from (1,1). Figuring out when/where these methods are applicable seems like an important result from this paper, and Section 3.4 regarding GEOS-Chem assumes that reasonable OSf can be obtained for Regime II. However, the results in Fig 1D and text In 564-566 indicate that non-negative OSf is not attainable using this method for these past campaigns, even in environments not affected by known interferences on the sulfate fragmentation (i.e. acidity and AN). Perhaps, the recommendation is that if aerosol is in one of the other 3 regimes, more work will be needed to understand the influence of H2SO4 or AN, but if the aerosol is in regime II, the method may or may not work, depending on how much OS, AS, and OA are present.

536. Add "estimated" or "calculated" before pH here and for other uses

550-566. Like the previous comment, this paragraph sets it up like this method will work for specific conditions (Regime II), but its conclusion is that it still gives large negative OSf values for Regime II, but for errors in calibrations, noise and "other factors."

620. This is either inconsistent with the definition of fH2SO4 in Eqns 1 and 3 or it is written in a misleading way. Please correct and be specific.

633. The pH estimation shows good correlation under certain conditions (pH estimated from E-AIM

707. How does one to assess whether the results for OSf are accurate for "Regime II" aerosol, even if they are non-negative, given all the interferences/complexity shown here?

**Technical Corrections:**

Organosulfate vs organic sulfate vs organic sulfur. Pick one or make distinctions clear (like it was done for MSA – that this is organosulfur, but not an organosulfate).

Define "Regime" I-IV the first time that they are mentioned with both pH and ANf and if they are redefined/reminding the pH and ANf, use consistent descriptions (e.g. Regime II: pH>0; ANf

533. Change "breaks down" to "is not applicable" or "does not quantify OSf"

570. Change % OS to OSf

Fig 3B. Typo in legend- "Troposphere"

Figure 3A. Points are very hard to distinguish from the background, particularly teal on gray, yellow on pink, yellow on blue, teal on green. I see these colors are used in Figure 4, so I understand the temptation to use them as background colors here, but the data points are almost impossible to see. Try dropping the background colors or find background/data colors with higher contrast.

632. Remove parentheses around "and potentially in time for a given instrument"

- 644. Remove parentheses before Guo
- 682. Redefine CE as collection efficiency for this section.

700. Change "chemical regime I in this work" to "Regime I"

Fig 6. Remove histogram from legend in 6A. Label B, C, D in descriptive terms. I.e. in C, Model -> GEOS-Chem

1069. Kang et al. Incomplete citation.

---

## Referee Comment (RC2) · Anonymous Referee #2 · 17 Dec 2020

This study examines the performance and validity of recently published OS estimation methods through analyzing the AMS spectral data of sulfate-related ions in ambient and lab-generated PM. This work reveals that the published OS estimation methods have major limitations and may produce erroneous results on OS concentration although could work under certain PM chemical composition regimes. In addition, this study explores the feasibility of estimating pH based on AMS spectral data and postulates the physical processes associated with sulfate fragmentation in the AMS. This exercise provides useful insights into why sulfate fragmentation changes in response to changes in aerosol chemical composition. This is a solid work that addresses an important topic related to atmospheric aerosol chemistry. This study is timely and sig-

nificant. The manuscript is well written and fits nicely within the scope of AMT. I thus fully support the publication of this work on AMT after the following comments are addressed.

It is mentioned that the fragment pattern of sulfate ions may vary from instrument to instrument or even for the same instrument after it is tuned. What's the range of variations in the fragmentation pattern of sulfate-related ions for inorganic sulfate?

It would be helpful to give a more clear definition of organosulfate (OS) here. The paper as it reads seems to refer OS to all organic compounds that can produce SOx ions. It is useful to note that not just the ROSO3 types of compounds generate SOx ions in the AMS, compounds with sulfone, sulfoxide, and sulfonate functional groups may do so as well.

For the PM data analyzed in this study, are there measurements other than the AMS that can be used to validate the quantification of OS concentration?

Consider to increase font size in the figures to make the texts more readable.

Line 41, fraction of what mass? Please clarify.

Line 46 – 47, this sentence is vague, what values of measured ammonium balance or HySOx/SOx ratio that are indicative of pH < 0?

Line 102, spell out SOAS

Line 147, what "sticky" means here, in what sense or towards what substrate?

Line 221-222, please elaborate a bit more on the "alternative methods" mentioned here

Line 264, remove "as".

Eq 6, stay consistent with the nomenclature, add square parenthesis to denote concentrations

Eq. 8, is it inorganic NO3 or total NO3? was the contribution of organic nitrate signals

removed?

Line 507, "in the absence of acidity" does not make sense.

Line 716, it would be interesting that the authors explain what "careful instrument calibration" involves, through analysis of pure inorganic sulfate particles?

---

## Author Comment (AC1) · 20 Jan 2021

Response to reviewers for the paper:

**Aerosol pH Indicator and Organosulfate Detectability from Aerosol Mass Spectrometry Measurements**

Schueneman et al., *AMTD*, 2020

We thank the reviewers for their useful comments on our paper and for their time spent reviewing it. To guide the review process we have copied the reviewer comments in black text. Our responses are in regular blue font. We have responded to all the referee comments and made alterations to our paper (**in bold text**). *Changes to text in figure captions are shown in bold italic text.*

**Anonymous Referee #1**

Summary:

This paper discusses the challenges of using AMS sulfate ion fragments to distinguish organic sulfates from inorganic sulfates and relates sulfate ion fragment ratios and ammonium balance to aerosol pH calculated from E-AIM. The authors apply previously published methods [Chen et al., 2019 and Song et al., 2019] to several past airborne AMS datasets to test the applicability of these methods for many aerosol compositions. They also determine sulfate ion fragment ratios for mixtures of ammonium sulfate (AS) C with inorganic and organic species in laboratory experiments. While these methods should be useful in some regions (pH>0, low ammonium nitrate - AN) based on the laboratory results presented here, the authors have determined that these methods often produce nonsensical values (negative OSf enrichment), suggesting that these methods cannot be used universally across all aerosol compositions. Specifically, high AN and high acidity change the way that sulfate interacts with the AMS vaporizer and prevents use of the Chen method in polluted (high AN) and remote marine regions (high acidity). However, even in the regions that should give good results (low AN, low acidity), the calculated organic sulfate fraction is often negative. The authors also show that sulfate ion fragments ratios correlate to E-AIM-calculated pH under certain conditions (pH<0, low AN), but correlations of measured ammonium balance with calculated pH are generally better.

General Comments:

As more people apply the methods described in Chen and Song to estimate organic sulfate from sulfate ion fragment ratios in a variety of atmospheric environments, the challenges to those methods highlighted in this paper will arise. The authors show that the Chen and Song methods are not always reliable for all aerosol compositions when applied to large AMS datasets accumulated over several campaigns spanning geographical locations, seasons, altitudes, and chemical regimes. They provide explanations as to how the sulfate ion fragment ratios are affected by high acidity, high AN, and somewhat by high organic aerosol concentrations at the

AMS vaporizer through laboratory experiments. This paper is of interest to many AMS users (and data users) as estimating organic sulfate fractions are applied to past and future datasets. Overall, this work is highly important and should be published after addressing the following comments below.

R1.1, This paper contains important results and discussion of interest to a large audience, but it would benefit from trimming and streamlining the text to improve readability. There is a lot of data and interesting analytical subtleties, but the paper frequently jumps around, especially when describing figures. For example, Fig. 1 is complex with a lot of data (lab plus field campaigns). Parts of this figure are described for the first time in disjointed places: Sect 3.1, 3.3 (but not 3.2), and 3.5.1. To help connect the laboratory and field applications of the Chen method, I suggest that the description and discussion of Fig. 1 be presented in its entirety and Section 3.2, which contains explanations for how the AMS responds to various species, could be reduced, moved, or integrated into the discussion of Figure 1.

We have revised the paper and streamlined the text where possible. We have kept what was Sect. 3.2 in the AMTD version (interpretation of sulfate fragmentation) and moved it to be Sect. 3.3 (after the section about applying the triangle method to field data). Regarding that topic and also Fig.1, we discussed internally before submission whether to break up Figure 1 into several figures, and went through several iterations and ordering of the sections. At the end we found that alternative configurations also had their own problems, given that the topics are quite complex and interrelated. Therefore we prefer to keep Fig. 1 as it is, which has the important advantage of allowing easy comparison of all the data in one place.

The new section 3.3 (3.2 in AMTD) contains very useful details for AMS / ACSM users who wish to understand the trends, and with the new organization (3.1: laboratory data, 3.2: aircraft data, and 3.3: interpretation) we believe it does not need to be trimmed. However, it does contain too much detail for others that are not very familiar with AMS details and may only be interested in using data provided by others. To clarify this and guide the flow of the paper for different readers, we have added the following text at the start of section 3.3:

**"We note that this section (3.3) should be of most interest for AMS/ACSM users, and can probably be skipped by others."**

Specific Comments:
R1.2, ln 58. Can you/ did you apply these methods to ACSM data? It seems that assumptions about hydrocarbon contributions at unit mass resolution, especially at m/z 81 ($HSO_3^+$, $C_6H_9^+$, $C_5H_5O^+$), would further increase uncertainty in this quantification. Just an interesting thought, since the ACSM was brought up here.

No, we did not apply these methods to ACSM data. We have added the following text to address this point:

**"We have not explored the application of these methods to ACSM data. ACSM data are unit-mass resolution, and the interferences between species at a given unit mass are estimated using a fragmentation table approach (Allan et al., 2004). This approach introduces more uncertainties, as exemplified by Hu et al. (2015) for similar fragment-based methods."**

R1.3, ln 76. Please check that these are the intended citations. I don't see organic sulfate fractions in Riva 2019a "Evaluating the performance of five different chemical ionization techniques for detecting gaseous oxygenated organic species," so perhaps that is not the correct reference.

You are correct, we mistakenly cited the wrong Riva et al. (2019). It has been corrected in the text and bibliography:

**"Until recently, most studies have shown that the OS molar fraction ($OS_f$ = OS / (AS + OS), calculated using only the sulfate moiety of the molecules) typically makes a small (~1-10%) contribution to total sulfate in $PM_1$ (e.g. (Tolocka and Turpin, 2012; Hu et al., 2015; Liao et al., 2015; Riva et al., 2016, 2019))."**

**Citation:"Riva, M., Chen, Y., Zhang, Y., Lei, Z., Olson, N. E., Boyer, H. C., Narayan, S., Yee, L. D., Green, H. S., Cui, T., Zhang, Z., Baumann, K., Fort, M., Edgerton, E., Budisulistiorini, S. H., Rose, C. A., Ribeiro, I. O., E Oliveira, R. L., Dos Santos, E. O., Machado, C. M. D., Szopa, S., Zhao, Y., Alves, E. G., de Sá, S. S., Hu, W., Knipping, E. M., Shaw, S. L., Duvoisin Junior, S., de Souza, R. A. F., Palm, B. B., Jimenez, J.-L., Glasius, M., Goldstein, A. H., Pye, H. O. T., Gold, A., Turpin, B. J., Vizuete, W., Martin, S. T., Thornton, J. A., Dutcher, C. S., Ault, A. P. and Surratt, J. D.: Increasing Isoprene Epoxydiol-to-Inorganic Sulfate Aerosol Ratio Results in Extensive Conversion of Inorganic Sulfate to Organosulfur Forms: Implications for Aerosol Physicochemical Properties, Environ. Sci. Technol., 53(15), 8682–8694, doi:10.1021/acs.est.9b01019, 2019."**

R1.4, ln 91. Instead of writing "inorganic" in quotations, be please specific: that these are ions that do not contain carbon and are typically categorized as /associated with inorganic sulfate. However, organic sulfate can also produce these ions.

We have revised the text to state:

**"The vaporization and ionization of AS and OS in the AMS produce similar ion fragments that do not contain a carbon atom, the major ones quantified being $SO^+$, $SO_2^+$, $SO_3^+$, $HSO_3^+$, and $H_2SO_4^+$. These ions were attributed primarily to inorganic sulfate in earlier AMS analyses (e.g. Jimenez et al. (2003)), but have been known to have a contribution from organosulfates since Farmer et al. (2010)."**

R1.5, ln 93. Decomposition or evaporation + ionization rather than ionization/decomposition? Section 3.2 and Fig 2C does a nice job of explicitly showing which species evaporate or thermally dissociate (and ionize) to form which ions, so perhaps use the same ideas and consistent language here and move parts or all of Fig 3 to the introduction.

See response to R1.1.

R1.6, Table 1. Be consistent. i.e. DC3: Summer 2012. ATom-1: Boreal Summer/ Austral Winter [sampling covered both northern and southern hemispheres at the same time but in different seasons, correct?

The table has been updated to read:

| Campaign | Location | Season/Year | References | Reference Label |
|---|---|---|---|---|
| DC3: Deep Convective Clouds and Chemistry | Mid-Latitude Continental United States | **Spring/Summer** 2012 | **Barth et al., (2015)** | A |
| SEAC$^4$RS: Studies of Emissions and Atmospheric Composition, Clouds and Climate Coupling by Regional Surveys | Continental United States | Summer 2013 | **Wagner et al., (2015); Toon et al., (2016)** | A, F, C |
| WINTER: Wintertime Investigation of Transport, Emissions, and Reactivity | Eastern United States, continental and marine | Winter 2015 | **Jaeglé et al., (2018); Schroder et al., (2018)** | A, F, C |
| KORUS-AQ: Korean United States Air Quality | South Korean Peninsula and Yellow Sea | Spring 2016 | **Nault et al., (2018)** | A, F, C |
| ATom-1: Atmospheric Tomography Mission 1 | Remote Pacific and Atlantic Basins | Boreal Summer/**Austral Winter** | **Brock (2019); Hodshire et al., (2019); Hodzic** | A, F, C |

| | | 2016 | et al., (2020) | |
|---|---|---|---|---|
| ATom-2: Atmospheric Tomography Mission 2 | Remote Pacific and Atlantic Basins | **Austral Summer**/Boreal Winter 2017 | **Hodzic et al., (2020)** | A, F, C |

R1.7, ln 266. Definition of HySOx: The distinction between its definition in Eq 5 (HySOx = H2SO4 + HSO3) and Eq 10 (HySOx = H2SO4 + HSO3+ SO3) is clear in the text, but it's annoying to have two definitions (and quantities) for the same variable in one paper. The distinction is lost when first introduced (total sulfate signal is defined on ln 96, but an undefined HySOx is used on ln 106).

We apologize for the confusing notation. This mistake occurred because Song et al. ((2019) and Chen et al. (2019) were defining the term $H_ySO_x^+$ differently. We have changed the notation when we describe the Song et al. method, as $H_ySO_x^{+,*}$ as shown here:

**$H_ySO_x^{+,*}$ (which differs from the notation used in Song's paper, but is necessary to differentiate $H_ySO_x^+$ between the Chen and Song papers)  is defined in Song et al. (2019) as $(SO_3^++HSO_3^++H_2SO_4^+)$."**

R1.8, ln 352. What altitude is this? This reference to the DC-8 seems out of context. Does this paragraph only address marine amines at minimum altitudes during ATom only?

The ATom campaigns took place on board the NASA DC-8, which is why it is mentioned here. Upon further inspection of Sorooshian et al. (2009), cited in that section, we have revised the text to say:

**"Another study found that amine mass concentration dropped off quickly with altitude to concentrations less than 25 ng m$^{-3}$ at an altitude between 200 and 300 m, which is the approximate minimum altitude flown on the DC-8 during the ATom campaigns (Sorooshian et al., 2009)."**

R1.9, ln 347-364. Consider moving this paragraph from the methods to discussion section

We originally had this section in the discussion but feel it is important to bring this topic up in the methods as it gives credence to the pH results we get from E-AIM. Also, several colleagues have asked us about the impact of amines on our pH calculations, which is another reason to discuss it in the methods to support the validity of the estimation. We also cannot find a location in the discussion section where this text would not be a distraction in the flow of the manuscript.

R1.10, ln 387. The thing that strikes me about Figure 1 is that even though "data are expected to lie inside the triangular region," few of the lab or field averages do! The AS lab cases that should show have fragment ratios of pure AS, do not, and this is explored. The field campaign averages also do not have fragment ratios consistent with calibration AS, and this is partially explored. Stating this discrepancy up front would improve readability of the discussion section.

We agree and have added another sentence to the end of that paragraph commenting on this observation (which we explore further in the following sections):

**"From applying this method, it is clear that none of the campaign averages or laboratory data falls between the [0,0] and [1,1] points, suggesting that there may be additional factors (other than sulfate composition) impacting the location of data in this triangular region."**

R1.11, ln 395. I think this example is unnecessary. If I'm understanding correctly, even if 50:50 OS:AS gives a non-negative OSf in the ANf= 0.95 case, the method would indicate there is no OS which is also not correct. Another way to say this is that as the endpoint for AS with high AN moves away from pure AS due to the interference from AN, the method estimates OSf with increasing inaccuracy so that it would predict OSf= 0%, when OSf is really 50% in this extreme case. So, even though the result (OSf = 0) is a "reasonable" non-negative OSf, it is still completely incorrect.

We have modified the explanation to follow the structure in the reviewer's comment, to enhance the readability of this section and provide increased clarity as to why the $OS_f$ method can be incorrect even in the situation where it provides a value > 0. We have changed the text as:

**"When $AN_f$ is increased past 0.50, there is an increase in both $nfH_ySO_x^+$ ions, even when all of the particulate sulfate is inorganic. As the particle $AN_f$ increases up to $AN_f$=0.95, the $OS_f$ estimation becomes increasingly inaccurate. The method may estimate $OS_f$=0% in the latter situation, when $OS_f$ is actually 50%. While $OS_f$=0% may be reasonable in some parts of the atmosphere, and one may be inclined to accept this result as it is non-negative, it is actually incorrect due to the effect of particulate AN."**

R1.12, ln 399. Figure 2b does not effectively show the information in this paragraph since the points overlap on a small region of the plot. What is the ratio of OA/AS when OA > 100 ug/m3? All other experiments are in relative terms, so 100 ug/m3 does not have any context here. How does OA/AS ratio compare to that of the campaign averages? The regime II averages?

We assume the reviewer is referring to Figure 1b, not 2b. We agree that all of the details of the $OA_f$ trend discussed in that paragraph may not be fully discernable in Fig. 1a. We have revised the text to also refer the reader to the specific panels in Figure S6 earlier and throughout this

discussion. Those figures clearly show all of the trends described for each dataset separately. However, we consider it informative to include the summary of that data as Fig. 1b for direct comparison to the other effects in the other panels, with all graphs on the same scale. The fact that for all experiments at modest OA concentrations $nfH_ySO_x^+$ ratios are only slightly altered is the most important observation for ambient data considerations. We have also listed the average $OA_f$ values for the two average OA>100 ug/m3 points on the Fig. 1 caption and stated the maximum OA values that are beyond the colorbar scale for the two terpene datasets in Figs. S6 a,c (in caption). Additionally we added a description of the 2014 terpene SOA trends which were not previously discussed.

The revised text for that paragraph and the figure captions now read:

R1.12a: Revised text (Sect. 3.1):

**"The effect of OA internally mixed with AS on the sulfate fragmentation pattern was also explored with toluene, alkanol, and monoterpene SOA (Fig. 1B and Fig. S6). For the alkanol SOA experiments we found that the presence of even a small coating of alkanol SOA (which is thought to be liquid (Liu et al., 2019)) shifts the normalized AS [1,1] point to ~[1.08,1.08], but increases in the fraction of OA ($OA_f$) from 0.1 to 0.3 lead to no further changes in $nfH_ySO_x^+$ (Fig. 1B). This means that for a sample containing a mixture of AS and alkanol SOA, the calculated $OS_f$ would be -15% (Chen). In contrast, toluene SOA, which spans $0 < OA_f < 0.5$, shows no clear change in the $nfH_ySO_x^+$ ions, indicating that $OA_f$ would not bias the Chen method for this example. The monoterpene SOA, from two different experimental datasets (2014 and 2015) using different AMSs, show more varied results than the previous two studies. Overall, the 2014 data shows a very small increase in the "pure" AS value in the $OA_f$ range 0-0.50, whereas the 2015 monoterpene data shows a consistent and constant 10-20% increase in $nfH_ySO_x^+$ compared to the pure AS calibration point (similarly to the alkanol SOA). However, when $OA_f$ is in the range of $0.50<OA_f<0.70$, 30-40% increases are observed for the 2014 and 2015 data. This result is only applicable to a few of the experiments (see Fig. S6), potentially due to very high SOA loadings (up to 300 µg m$^{-3}$). These high OA concentrations could potentially lead to a change of the particle phase due to condensation of more volatile and liquid species, potentially altering the interactions of the particles and the vaporizer surfaces. These experiments collectively suggest that a "pure" AS calibration point of [1.15,1.15] may be more appropriate when applying the Chen et al. method to some mixed aerosol at typical OA concentrations observed in the atmosphere; this is discussed further in Sect. 3.2."**

R1.12b: Figure 1 caption:
***"Fig. 1. Laboratory and field data for sulfate fragmentation shown in the triangle diagram proposed by Chen et al. (2019). (A) Data split into 10 quantiles of $AN_f$ value for the full***

*KORUS-AQ campaign, as well as for different laboratory internal mixtures of AS and AN .
(B) Data from two chamber experiments, split into 5 quantiles of $OA_f$. Data with very high OA
(>100 µg m$^{-3}$) are shown as grey triangles. The average of $OA_f$ for the very high OA data in
2014 and 2015 is 0.8.Two separate datasets of monoterpene SOA chamber experiments are
labelled as "2014" and "2015". (C) Data split into 10 quantiles by calculated pH for ATom-1
and 2, colored by calculated pH from E-AIM. (D) Averages for 5 aircraft campaigns for the
full campaign and a subset of each campaign where pH<0 and $AN_f$ <0.3."*

R1.12c: Figure S6 caption:

*"Fig. S6. $fH_ySO_x^+$ ions vs. $OA_f$ colored by total OA concentration (left and bottom right) and
top right colored by experiment index. All data are from chamber experiments where SOA was
formed on ammonium sulfate seed aerosol from (A, B, C, D) nitrate radical reaction with
monoterpenes (where 2014, 2015 represent different series of experiments done in different
years and different instruments), and photooxidation of (E) alkanols and (F) toluene. The
maximum [OA] concentrations observed in (A) and (C) are 204 and 206 µg m$^{-3}$, respectively.
The $fH_ySO_x^+$ ratios have been normalized to the average ratios for the ammonium sulfate seed
for each experimental dataset."*

R1.12d: We also added another paragraph and figure (Fig. S7) to Sect. 3.2 and the SI, with text
stated as such:

**"The effect of OA (shown in Fig. 1B for laboratory data) on sulfate fragmentation in
ambient data is less clear due to the lack of data that has a lower $AN_f$, higher pH, and
little/no OS (see Table S1 for average campaign $OA_f$). In the presence of any one of those
factors, the sulfate fragmentation will be affected. It is especially challenging to confirm the
absence of OS, due to the lack of direct total OS measurements available. In Fig. S7, we
isolate a subset of the KORUS-AQ dataset (where $AN_f$<0.3 and pH>0, defined as "regime
II" and discussed in detail in Sect. 3.4) to see if there is an offset in the AS under these
chemical conditions as observed in the laboratory data shown in Fig. 1B. Similarly to the
lab data, there appears to be a ~10% offset between the pure AS $fH_ySO_x^+$ values from
calibrations, and the KORUS-AQ data that occupies regime II (average $OA_f$~43%). This
offset is smaller than some of the offsets observed in from the laboratory data (Fig. 1B and
S6), but may hinder the ability of the Chen $OS_f$ quantification method to estimate [OS]
even in conditions where the pH>0 and the $AN_f$<0.3."**

R1.12e: SI Figure S7 and caption:

[Figure]

*“Fig. S7. fH$_y$SO$_x$$^+$ ions (not normalized) vs. time for AS calibrations and ambient sampling for the KORUS-AQ campaign. (A) shows fH$_2$SO$_4$$^+$ and (B) shows fHSO$_3$$^+$. The yellow line shows smoothed, average data for the campaign, and the grey points show the non-averaged ambient data. Black triangles show the average fH$_y$SO$_x$$^+$ values for four pure AS calibrations done during KORUS-AQ.”*

R1.13, ln 415. The hypothesis here is that you can change the relative HxSOy fragments due to different residence times on the vaporizer for OA-coated particles (i.e. less bounce, more thermal decomposition)? Update: I see this gets discussed more in Section 3.2, so perhaps it would be better to introduce these ideas within Section 3.1.

See response to R1.1.

We have modified the text to make an explicit reference to section 3.2 (now 3.3) for interested readers:

**“These high OA concentrations could potentially lead to a change of the particle phase due to condensation of more volatile and liquid species, potentially altering the interactions of the particles and the vaporizer surfaces (see section 3.3).”**

R1.14, ln 425. I suggest putting Section 3.2 somewhere else. Before 3.1? After 3.3? Figure in the SI with the key points in the discussion of Fig 1? Moving the generalize fragment discussion (Fig 3C) to the intro? Figure 1 includes both lab and field data, and it seems odd that the discussion of a single figure is broken up like this without any indication that the discussion will be picked up again. I got to the end of Section 3.1 wondering about Fig 1D and why all the campaign averages are out still of the "Chen triangle." It seems to me that Section 3.2 contains the analytical crux of why this method isn't universally applicable, so this is an important section. It just seems in a weird place. This discussion of the AMS response is applicable for both the lab and field data, so I think it makes sense to describe Figure 1 in it's entirety, and then, as a discussion, talk about

the fragmentation for these different conditions (in context of both the lab experiments and the complex mixtures in the ambient data).

See the responses to R.1.1. After having tried alternative ordering of the material, we agree that the order suggested by the reviewer would be less confusing. We now describe all of Fig. 1 in Sect. 3.1 and Sect. 3.2, and then describe Fig. 2 in Sect. 3.3.

R1.15, ln 458. Based on this explanation, that particles with longer contact time with the vaporizer experience higher temperatures and thus more thermal decomposition, I would expect the sulfate ion fragment ratios to depend on the vaporizer temperature. Has sensitivity to vaporizer temperature been tested?

This is a good point. Relevant data were included in a previous study, and we have added the following text to refer to them here:

**"Consistent with this interpretation, it was shown that the $H_2SO_4^+/SO^+$ fragment ratio increased as the vaporizer temperature was reduced while sampling ambient air, while the $SO_2^+/SO^+$ ratio did not change (Docherty et al. (2015), their figure S5)."**

R1.16, ln 506. Define pH>0 and AN<0.3 as Regime II here in the text. Combine this paragraph with ln 526-533.

We do not agree that line 506 and the paragraph in lines 526-533 should be combined, as this could potentially create confusion. In this text we first introduce Fig. 1D and say what we observe, then go through it by discussing ATom and KORUS-AQ campaigns, and finally explaining the additional points from the SEAC⁴RS and WINTER campaigns. We did change the text to specify regime I:

**"In Fig. 1D, average values for each campaign in regime I, defined as $AN_f < 0.3$ and calculated pH > 0, are shown."**

R1.17, ln 533. This is a general comment: I'm struggling with the paper setting up the discussion that even though the proposed method does not work for high acidity or high AN as shown in the lab results, it will work for subsets of ambient data that meet the criteria (pH>0; AN<0.3). However, in the end, the method does not work for any subsets of the ambient data as shown in Fig. 1D. This is partially explained by organic aerosol moving the AS fragment ratio away from (1,1). Figuring out when/where these methods are applicable seems like an important result from this paper, and Section 3.4 regarding GEOS-Chem assumes that reasonable OSf can be obtained for Regime II. However, the results in Fig 1D and text ln 564-566 indicate that non-negative OSf is not attainable using this method for these past campaigns, even in environments not affected

by known interferences on the sulfate fragmentation (i.e. acidity and AN). Perhaps, the recommendation is that if aerosol is in one of the other 3 regimes, more work will be needed to understand the influence of H2SO4 or AN, but if the aerosol is in regime II, the method may or may not work, depending on how much OS, AS, and OA are present.

We have revised the text to make clearer that even when considering only the data in regime II, that appears unaffected by $AN_f$ and pH, the $OS_f$ method still failed for most of our campaigns. The only way to truly know if the OS estimation method works is to have a completely independent measurement of total OS. An approximation to total OS was only available to us for ATom-1 and -2, and we comment on this on the revised text. We have altered the wording of Sect. 3.4, in reference to regime II, to make it clear that this method did not work for our datasets:

**"Regime II (lower right) involves less acidic conditions (calculated pH > 0) and lower $AN_f$ (< 0.3). In this region sulfate fragmentation in the AMS is not strongly impacted by either $AN_f$ or acidity. In principle, in this regime the recently proposed sulfate deconvolution methods could be applicable. The geographical regions studied in Chen et al. (2019) and Song et al. (2019) generally fall in this regime, and this might explain the lack of large negative $OS_f$ values in those studies, in contrast to our observations for other regions. About half of our campaign data is located in this regime, more so for the continental campaigns and much less so for the remote campaigns. Specifically, 65% of KORUS-AQ, 60% of DC3, 87% of SEAC[4]RS, 11% of ATom-1 and 20% of ATom-2 fall in this regime. We have applied the 1D version of the Chen method to each field campaign after filtering it by the $AN_f$ and calculated pH constraints for regime II. $OS_f$ is nominally slightly greater than zero for ATom-1, $OS_f \sim 3\%$, of the order of the 0.3% estimate in regime II from PALMS (for ATom-1 and ATom-2, estimated by only considering the sulfate moiety from the IEPOX or glycolic acid sulfate (GAS) OS, neither of which was detected in the supermicron aerosol (Froyd et al., 2009, 2019; Liao et al., 2015)(see Fig. S7). However, $OS_f$ is much less than zero for ATom-2 ($OS_f \sim -23\%$) and KORUS-AQ ($OS_f \sim -26\%$). These unreasonable results may be due to the effect of OA on sulfate fragmentation in the AMS (discussed in Sect. 3.2). For this reason, strong caution is advised in applying $OS_f$ estimation methods to ambient data, even in regime II. In addition, estimating OS with sulfate ions may be susceptible to errors due to inaccuracies in AS calibrations, noise present in the ambient data, or other factors."**

New abstract figure:

[Figure]

R1.18, ln 536. Add "estimated" or "calculated" before pH here and for other uses

We have clarified this in the methods section with the text below as (and modified the paper accordingly):

"**We have added the modifier "calculated" before pH for all situations where we are describing the E-AIM pH, and "estimated" when we refer to pH from the empirical estimation methods from AMS measurements, introduced in this study."**

R1.19, ln 550-566. Like the previous comment, this paragraph sets it up like this method will work for specific conditions (Regime II), but its conclusion is that it still gives large negative OSf values for Regime II, but for errors in calibrations, noise and "other factors."

See the response to comment R1.17.

R1.20, ln 620. This is either inconsistent with the definition of fH2SO4 in Eqns 1 and 3 or it is written in a misleading way. Please correct and be specific.

We accidentally wrote our new variable, $H_ySO_x^+/SO_x^+$ instead of the $fH_ySO_x$ values. This has been corrected:

**"Fig. 1C shows that $fH_2SO_4^+$ and $fHSO_3^+$, i.e. the amount of sulfate fragments retaining one or two hydrogens ($H_2SO_4^+$ and $HSO_3^+$) relative to the total sulfate fragments ($H_2SO_4^+$, $HSO_3^+$, $SO_3^+$, $SO_2^+$, and $SO^+$) increases as calculated pH decreases."**

R1.21, ln 633. The pH estimation shows good correlation under certain conditions (pH estimated from E-AIM <0), so I'm thinking the real-time estimate of pH from ammonium balance and HySOx/SOx is useful only if you already know that you are in a highly acidic environment through an independent estimation, as indicated here. Is there an alternative filter than E-AIM pH<0 for the relevant conditions?

This is not correct for the better method (ammonium balance), as the method itself allows you to know whether you are in a part of the atmosphere when the method works, i.e. where ammonium balance < 0.65. Another rule of thumb is to look at data that is exclusively outside of the boundary layer. This is mentioned already in the conclusions. We have also added a brief statement to Sect. 3.5.2:

**"As shown in Fig. 6, $NH_{4\_bal}$ and calculated pH for the aircraft studies show a strong and consistent relationship in regime I (calculated pH < 0), providing another potential method for estimating pH (all one needs to use this method is the ammonium balance, and if it is < 0.65, the method should be applicable)."**

The reviewer is correct for the fragment-based pH estimation method, but in any case that method is clearly noisier and less desirable.

R1.22, Fig 5. Is just Regime I data shown and used for the ATom fit? (i.e. E-AIM pH<0 only for ATom)

Figure 5 shows ATom and KORUS-AQ data, but only the data where pH<0 is fit. When pH<0, the AN partitions out of the particle phase, so this is exclusively regime I data being fit. We added the following text to the Fig. 5 caption:

***"when calculated pH<0"***

R1.23, Fig 7. pH estimated by nHySOx/SOx is missing for a large portion of this time series, but there seems to be more data points in Fig 7D. Is this a data gap in the sampling, or an issue with the application the pH estimation method?

You are correct, for Fig. 7B some of the data is missing for $nH_ySO_x^+/SO_x^+$ estimated pH. This is because of slow evaporation of sulfate in the AMS in the second half of the transect, leading to altered fragmentation. The Fig. 7 caption has been altered to mention this point as:

*"(B) Time series of sulfate and pH for a large power plant plume sampled during WINTER, only a few data points are shown for pH estimated from $nH_ySO_x^+/SO_x^+$ because sulfate in the AMS evaporated slowly during the second half of the plume transect, leading to altered sulfate fragmentation, and this effect cannot be corrected for, due to infrequent backgrounds in aircraft fast acquisition mode."*

In Fig. 7D, we do not show the $nH_ySO_x^+/SO_x^+$ estimated pH results for the $SO_2$ plume during WINTER because there are only a few data points.

R1.24, ln 707. How does one to assess whether the results for OSf are accurate for "Regime II" aerosol, even if they are non-negative, given all the interferences/complexity shown here?

See response to R1.17.

A recent paper in ES&T titled "Overestimation of monoterpene organosulfate abundance in aerosol particles by sampling in presence of $SO_2$" also mentions that even other OS measurements in past literature may have substantial associated positive biases (Brüggemann et al., 2020).

We have added this sentence to the introduction:

**"It should be noted that a recent study reports that OS filter-based measurements in past scientific studies may have substantial associated positive biases, leading to an overestimate for [OS] (Brüggemann et al., 2020)."**

We have added this sentence to the conclusions to further emphasize this point:

**"For the ambient data analyzed here, even in regime II the $OS_f$ estimation produced nonsensical results. Extreme caution is recommended to anyone who chooses to apply the $OS_f$ estimation methods."**

Technical Corrections:

R1.25, Organosulfate vs organic sulfate vs organic sulfur. Pick one or make distinctions clear (like it was done for MSA – that this is organosulfur, but not an organosulfate).

We do not see anywhere where we used the term "organic sulfur" but we have changed any mention of "organic sulfate" to **"organosulfate"**, for consistency.

R1.26, Define "Regime" I-IV the first time that they are mentioned with both pH and ANf and if they are redefined/reminding the pH and ANf, use consistent descriptions (e.g. Regime II: pH>0; ANf<0.3)

See R1.16 for regime I definition (added to the text at the first place regime I was mentioned).

We also define Regime II in R1.12d.

Sect. 3.4 defines all of the regimes in detail. We tried adding this to the abstract, and it did not fit or flow well.

R1.27, ln 79. Missing source of sulfate production?

We modified this text to read:

**"Another important recent subject of debate is the missing sources of sulfate production in haze events in China..."**

R1.28, ln 89. "Recent AMS work." Reference?

We added two references: Chen et al. (2019) and Song et al. (2019).

R1.29, ln 95. "Recent laboratory studies" but only one reference

It is only one reference, Chen et al. (2019), where they conducted multiple laboratory studies. Text was changed for clarity:

**"However, a recent laboratory study with many OS standards found reproducible differences in the fragmentation of AS vs OS (Chen et al., 2019)."**

R1.30, ln 107. Delete "based on their method"

The text now states:

**"These authors reported $OS_f$~17%±7% (which corresponds to [OS] ~ 5-10 µg m$^{-3}$) during winter haze episodes in China."**

R1.31, ln 264. delete "as"

We deleted "as".

R1.32, ln 310. Add model numbers of DMA and CPC in SMPS to be consistent with level of detail in previous paragraph.

The text now states:

**"Aerosol composition was monitored by AMS and size distributions were monitored with a scanning mobility particle sizer (SMPS: DMA was TSI Model 3081, electrostatic classifier was Model 3080, and the CPC was Model 3775)."**

R1.33, ln 319. total inorganic nitrate?

It is total nitrate (organic+inorganic) which was shown in Nault et al. (2020) to not impact the calculated pH or ammonium balance. The text was modified to include this:

**"Total nitrate (inorganic+organic) was input, as Nault et al. (2021) found that removing estimated organic nitrate does not significantly impact the pH calculation."**

R1.34, ln 318/325. Consolidate sentences. Both describe the same model input.

We consolidated the sentence mentioned by the reviewer, and the paragraph now states:

**"Aerosol pH was estimated using the Extended Aerosol Inorganic Model (E-AIM) Model IV (Clegg et al., 1998; Massucci et al., 1999; Wexler and Clegg, 2002). We input into the model (ran in "forward mode"), the total nitrate (gas-phase $HNO_3$ plus particle-phase total $NO_3^-$), sulfate, ammonium, relative humidity (calculated according to the parameterization of Murphy and Koop (2005), which is critical for upper tropospheric conditions), and temperature. Total nitrate (inorganic+organic) was input, as Nault et al. (2021) found that removing estimated organic nitrate does not impact the pH calculation (Nault et al., 2021). This was done to calculate aerosol liquid water and aerosol pH. Model IV was not run with chloride ions, as their concentrations were very low, and including chloride limits the model to temperatures $\geq 263$ K (Friese and Ebel, 2010), which would greatly limit the analysis of calculated pH for WINTER, ATom-1, and ATom-2. We have added the modifier "calculated" before pH for all situations where we are describing the E-AIM pH, and "estimated" when we refer to pH from the empirical estimation methods from AMS measurements, introduced in this study. Also, including chloride precludes running the model under supersaturated solution conditions, which is a closer approximation of ambient aerosol (Pye et al., 2019). All aerosol mass concentrations were from the CU AMS. $HNO_3(g)$ was measured by the California Institute of Technology chemical ionization mass spectrometer (CIT-CIMS) (Crounse et al., 2006), which was flown in all of these missions**

**(excluding WINTER, where the UW-CIMS was used for the HNO$_3$ measurements) (Lee et al., 2014, 2018).”**

R1.35, ln 320. To keep consistent tense in model description, change "is" to "was".

See R1.34.

R1.36, ln 332. Subscript HNO3

This has been done.

R1.37, ln 416. Remove paragraph break.

This has been done.

R1.38, ln 496. Delete D in "Fig. 1D." Field data is in Fig 1A, C, D.

Fixed to say:

**"The results of applying the Chen et al. (2019) method to five aircraft campaigns are shown in Fig. 1."**

R1.39, ln 533. Change "breaks down" to "is not applicable" or "does not quantify OSf"

This text was changed to:

**"It is observed that every single campaign average falls outside of the triangle (for the full campaign and non-acidic, low AN$_f$ averages), indicating that the Chen et al. (2019) method, as proposed, is not applicable to many regions of the atmosphere."**

R1.40, ln 570. Change % OS to OSf

Fixed to say:

**"For the entire atmosphere, shown in Fig. S10, the distribution for OS$_f$ looks similar to Fig. S9."**

R1.41, Fig 3B. Typo in legend- "Troposphere"

This typo was fixed.

R1.42, Figure 3A. Points are very hard to distinguish from the background, particularly teal on gray, yellow on pink, yellow on blue, teal on green. I see these colors are used in Figure 4, so I understand the temptation to use them as background colors here, but the data points are almost impossible to see. Try dropping the background colors or find background/data colors with higher contrast.

We explored different options, and we decided to darken the points in (A) for KORUS-AQ and DC3 and in (B) we increased the thickness of the markers. This was the best option to make them easier to see, while maintaining consistency of the color scheme across the manuscript.

[Figure]

R1.43, ln 632. Remove parentheses around "and potentially in time for a given instrument"

The parentheses have been removed.

R1.44, ln 644. Remove parentheses before Guo

The text says:

**"e.g., Guo et al., (2015), (2016); Hennigan et al., (2015); Weber et al., (2016)…"**

R1.45, ln 682. Redefine CE as collection efficiency for this section.

The text now states:

**"3.6 Possibility of Estimating Collection Efficiency (CE) from Sulfate Fragmentation"**

R1.46, ln 700. Change "chemical regime I in this work" to "Regime I"

The text now states:

**"We show using both laboratory and field data that both high acidity (regime I) and high AN$_f$ (regime III) result in major changes in sulfate fragmentation, which often lead to nonsensical results for the OS$_f$ methods."**

R1.47, Fig 6. Remove histogram from legend in 6A. Label B, C, D in descriptive terms. I.e. in C, Model -> GEOS-Chem

This was fixed. The revised figure is shown below.

[Figure]

R1.48, ln 1069. Kang et al. Incomplete citation.

Kang et al., (2016) has been fixed to say:

**"Kang, H., Day, D. A., Krechmer, J. E., Ayres, B. R., Keehan, N. I., Thompson, S. L., Hu, W., Campuzano-Jost, P., Schroder, J. C., Stark, H., Ranney, A., Ziemann, P., Zarzana, K. J., Wild, R. J., Dubé, W., Brown, S. S., Fry, J. and Jimenez, J. L.: A33E-0280: Secondary organic aerosol mass yields from the dark NO3 oxidation of α-pinene and-carene: effect of RO2 radical fate, in American Geophysical Union Fall Meeting., San Francisco, CA, USA, 12-16 December 2016."**
* * *
**Anonymous Referee #2**

R2.0. Summary: This study examines the performance and validity of recently published OS estimation methods through analyzing the AMS spectral data of sulfate-related ions in ambient and lab-generated PM. This work reveals that the published OS estimation methods have major limitations and may produce erroneous results on OS concentration although could work under certain PM chemical composition regimes. In addition, this study explores the feasibility of estimating pH based on AMS spectral data and postulates the physical processes associated with sulfate fragmentation in the AMS. This exercise provides useful insights into why sulfate fragmentation changes in response to changes in aerosol chemical composition. This is a solid work that addresses an important topic related to atmospheric aerosol chemistry. This study is timely and significant. The manuscript is well written and fits nicely within the scope of AMT. I thus fully support the publication of this work on AMT after the following comments are addressed.

R2.1, It is mentioned that the fragment pattern of sulfate ions may vary from instrument to instrument or even for the same instrument after it is tuned. What's the range of variations in the fragmentation pattern of sulfate-related ions for inorganic sulfate?

This was already shown in the AMTD version (Fig. S3). The range in $fH_2SO_4^+$ for pure ammonium sulfate is ~0.03-0.08, and $fHSO_3^+$ is ~0.04-0.16. These are the average values for pure AS calibrations during five campaigns.

This was already in our manuscript published in AMTD, discussed in Sect. 2.4, quoted below for clarity:

"All variables were normalized to the values of the same variables for pure AS calibrations (conducted during each field experiment) in order to eliminate some of the spread in the sulfate ions that is likely due to instrument-to-instrument or instrument-in-time variability (Fry et al., 2013; Chen et al., 2019) (Fig. S3)."

R2.2, It would be helpful to give a more clear definition of organosulfate (OS) here. The paper as it reads seems to refer OS to all organic compounds that can produce SOx ions. It is useful to note that not just the ROSO3 types of compounds generate SOx ions in the AMS, compounds with sulfone, sulfoxide, and sulfonate functional groups may do so as well.

We have added the following text to the introduction to clarify this point:

**"In contrast to nitrates, deconvolving inorganic vs. organosulfates (OS, which includes sulfonic acids, when present) is thought to be more difficult. The fragmentation pattern for one atmospherically relevant OS was similar to those of inorganic sulfates (mainly ammonium-sulfate salts, AS) in an early study, with minimal C-S-containing fragments (Farmer et al., 2010)."**

R2.3, For the PM data analyzed in this study, are there measurements other than the AMS that can be used to validate the quantification of OS concentration?

There are no measurements from these campaigns that fully quantify OS concentrations. The PALMS instrument can see major OS species (like glycolic acid sulfate and IEPOX-sulfate), see Fig. S7 for a visual comparison, as well as our discussion in Sect. 3.4.

R2.4, Consider to increase font size in the figures to make the texts more readable.

The font size in Fig. 3 (see R1.41) and Fig. 6 (see R1.47) were increased by one point. We think the current font size is sufficient in other figures and that it matches the 12 point font of the paper well. We will pay attention to the proofs of the final paper, as the sizing of the figures in that version can lead to problems as well.

Specific Comments:
R2.5, ln 41, fraction of what mass? Please clarify.

We have updated this text to read:

**"In regions with lower acidity (pH>0) and ammonium nitrate (fraction of total mass<0.3), the proposed OS methods might be more reliable, although application of these methods often produced nonsensical results."**

R2.6, ln 46 – 47, this sentence is vague, what values of measured ammonium balance or HySOx/SOx ratio that are indicative of pH < 0?

The text now states:

**"Under highly acidic conditions (when calculated pH<0 and ammonium balance<0.65), sulfate fragment ratios show a clear relationship with acidity."**

R2.7, ln 102, spell out SOAS

The text now states:

**"From this method, an average OS mass concentration ($C_{OS}$) of 0.12 μg m$^{-3}$ was estimated for the Southern Oxidant and Aerosol Study (SOAS) ground campaign in rural Alabama (Carlton et al., 2018), with $OS_f \sim 4\%$ (Chen et al., 2019)."**

R2.8, ln 147, what "sticky" means here, in what sense or towards what substrate?

The text has been modified for clarity as:

**"Many field studies do not include measurements of $NH_3$ or $HNO_3$, two species that are difficult to measure due to inlet delays caused by strong interactions with surfaces. Both species are typically present at low concentrations and not routinely measured, limiting the ability to calculate aerosol pH (Hennigan et al., 2015)."**

R2.9, ln 221-222, please elaborate a bit more on the "alternative methods" mentioned here

We apologize that the original sentence was confusing as written. What we meant to say is that if someone could come up with an alternative CE estimation method, that would be of interest. So we explore that (briefly) in Sect. 3.6. We added the text below for additional clarification:

**"Alternative methods to estimate ambient CE for ambient particles are of interest, we explore a potential alternative method here."**

R2.10, ln 264, remove "as".

We fixed it to say:

**"We also define a new AMS sulfate ion ratio, $H_ySO_x^+/SO_x^+$, and create the normalized $nH_ySO_x^+/SO_x^+$ to reduce the influence of instrument-to-instrument or instrument-in-time variability:"**

R2.11, Eq 6, stay consistent with the nomenclature, add square parenthesis to denote concentrations

In Eq. 5 we define $H_ySO_x^+/SO_x^+$ without using brackets, so that is why we do not use brackets on the left or right side of Eq. 6. We are only using the concentration brackets when we are denoting a value directly measured by the instrument.

R2.12, Eq. 8, is it inorganic NO3 or total NO3? was the contribution of organic nitrate signals removed?

The equation shown in the AMTD article states inorganic nitrate. We removed the estimated contribution of organic nitrate.

R2.13, ln 507, "in the absence of acidity" does not make sense.

We changed this to say:

**"For less acidic aerosols/conditions and in the absence of OS or $AN_f$ effects, it is expected that the data would fall on top of the [1,1] pure AS point in the 1D triangle plot, but this is not observed."**

R2.14, ln 716, it would be interesting that the authors explain what "careful instrument calibration" involves, through analysis of pure inorganic sulfate particles?

A paper on this topic is being worked on by our group that will discuss calibrations in more detail.

References:

[revised manuscript text omitted]